# Evolutionary Context Search for Skill Acquisition

## Abstract

Large Language Models struggle to reliably acquire new knowledge post-deployment—even when relevant text resources exist, models fail to transform them into actionable knowledge without retraining. Retrieval-Augmented Generation attempts to bridge this gap by surfacing relevant documents at inference time, yet similarity-based retrieval often fails to identify context that actually improves task performance. We introduce Evolutionary Context Search (ECS), an evolutionary method that searches over context combinations guided accuracy on a small development set, requiring only inference calls without weight updates. ECS moves beyond semantic similarity to discover non-obvious context pairings that significantly boost performance. On BackendBench, ECS improves correctness by 27% over the strongest RAG baseline, and it also achieves promising results on $\tau^2$-Bench and Terminal-Bench. Contexts evolved with Gemini-3-Flash transfer to Claude and DeepSeek, with transfer gains varying by model and benchmark. These results position ECS as a practical approach to post-deployment skill acquisition via automated discovery of effective, reusable contexts.

## 1 Introduction

Updating the knowledge of Large Language Models (LLMs) to acquire new capabilities after the training cutoff remains a technical challenge (Onoe et al., 2023; Zhong et al., 2023; Yao et al., 2023; Li et al., 2023b). Domain-Specific Languages (DSLs) like CuTeDSL for GPU programming have comprehensive documentation, yet adapting LLMs to write code correctly in such high-resource but unseen languages cannot be done reliably (Kandpal et al., 2023; Gu et al., 2025). The core problem is not missing information, but effective methods to harness novel and diverse information sources to efficiently adapt the pretrained knowledge of an LLM to acquire the new target capability.

Existing approaches to skill acquisition incur substantial computational costs while struggling to obtain the required skill. Training-based methods, such as supervised finetuning (SFT) and reinforcement learning (RL) on curated data, are expensive due to their computational requirements, with additional engineering costs incurred by data collection and processing (Cottier et al., 2024). Moreover, given post-training requires weight access, such methods are naturally inapplicable to frontier, closed-source models. Current in-context approaches offer only partial solutions to training-based methods. Retrieval-Augmented Generation (RAG) (Lewis et al., 2020; Ram et al., 2023; Khandelwal et al., 2019) can equip base models with new knowledge at test-time without necessitating weight access, but similarity-based retrieval often fails because queries tend to be verbose, contain irrelevant context, or not task-specific (Li et al., 2023a; Petroni et al., 2020; Yoran et al., 2023). More generally, RAG is highly sensitive to arbitrary context ordering and requires considerable human engineering effort (Akkiraju et al., 2024), impairing the efficacy and viability of the technique.

In this work, we leverage the viewpoint that text-based prompt augmentations offer a flexible and effective framework for updating an LLM's knowledge, and develop a search method for accumulating the required context that avoids the issues of retrieval entirely. Crucially, rather than relying on LLMs as evolutionary operators, we employ a simple genetic algorithm-style (Katoch et al., 2021) approach to evolve context combinations. We find this process to be highly efficient, typically requiring as few as 5 iterations to converge on high-utility results. This search process intuitively mirrors how humans learn new skills – collection of relevant documentation and other resources, assessing the fitness of a given resource by how it improves our

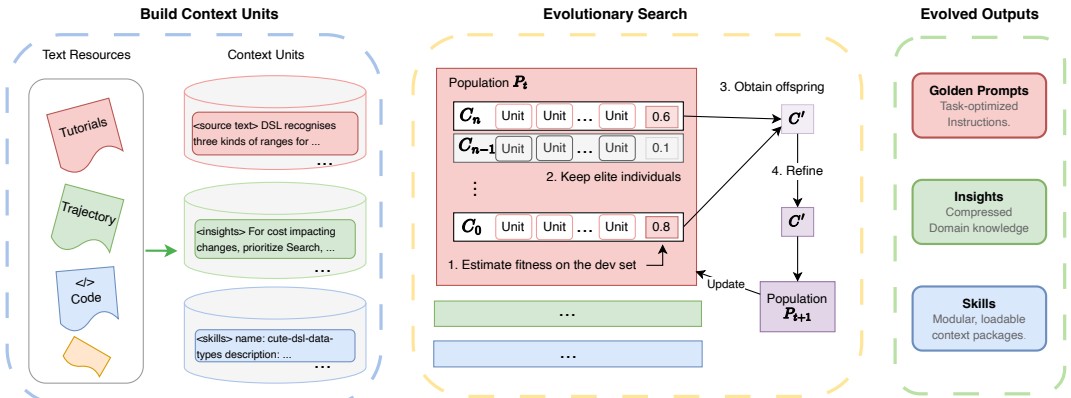

Figure 1: **Evolutionary Context Search.** Our method takes a population of text resources and evolves optimized contexts that confer the knowledge required to perform tasks in unseen domains. Each successive generation accumulates task-dependent, knowledge-rich information, effectively searching the corpora to obtain token-efficient contexts that enable novel skill acquisition in LLMs.

understanding, using such resources to discover followup works, and so on recursively until we accumulate the required contextual information to attain the new capability.

Building on these insights, we introduce Evolutionary Context Search (ECS) (Figure 1), a framework that extracts knowledge from provided corpora through fitness-guided evolution rather than similarity-based retrieval. ECS takes a context-centric view: ECS converts text resources into context units and combines them into candidate contexts. These contexts are then evolved to improve performance on specific tasks. The evolved context can take multiple forms: raw documentation, condensed summaries/insights, or structured agent skills (Anthropic, 2025b). This approach transforms passive text resources into active teaching materials that progressively improve model performance. Our approach provides three connected benefits: 1)performance gains over the evaluated baselines through task-driven context selection, 2) robustness to non-task-specific, verbose, or misleading queries that can impair retrieval, and 3) deployment without weight access or training.

Our contributions are three-fold:

1. We propose Evolutionary Context Search (ECS), a framework that treats context selection as an optimization problem to maximize skill and knowledge acquisition from external resources.

2. We empirically demonstrate that ECS provides substantial improvements over baselines across diverse benchmarks, including BackendBench, $\tau^2$-Bench, and Terminal-Bench.

3. We evaluate reuse of contexts discovered with Gemini-3-Flash on two held-out target models, finding positive BackendBench transfer and model-dependent results on $\tau^2$-Bench.

## 2 Related Work

**Knowledge updating.** Existing approaches to update an LLM's knowledge to unseen information can be broken into two sets, gradient-based and in-context. Gradient-based approaches include supervised fine tuning and reinforcement learning from curated data, which incur persistent challenges due to catastrophic forgetting (Luo et al., 2025; Shi et al., 2025), not to mention substantial engineering and computational overhead caused by training and data curation. Parameter efficient fine tuning methods (Hu et al., 2022; Dettmers et al., 2023; Tian et al., 2024) typically trade computational overhead for performance (Chen et al., 2022; Biderman et al., 2024), while still requiring expensive data collection. Additionally, gradient based methods require model weight access, thereby precluding application to powerful closed-source models. In-context approaches, on

the other hand, resolve the need for weight access. Retrieval augmented generation (Lewis et al., 2020; Ram et al., 2023; Khandelwal et al., 2019) aims to equip the model with new knowledge through similarity based indexing, but is highly sensitive to arbitrary factors such as ordering (Liu et al., 2024a). Our approach builds off of in-context instruction provision, but automatically discovers populations of resources to optimally confer the required knowledge update without any manual, iterative prompt refinement.

**Prompting.** Optimal performance in specialized tasks is highly dependent on the prompting strategy (Zhou et al., 2022; Wang et al., 2022; 2023; Madaan et al., 2023). Recent studies avoid manual prompt engineering by learning prompts through optimizing continuous embeddings Li et al. (2024); Sinhababu et al. (2025); Liu et al. (2024b); Qin & Eisner (2021); Lester et al. (2021). Evolutionary methods have emerged as powerful alternatives for prompt optimization which avoid the need for gradients. Agrawal et al. (2025) learn prompts through sampling system-level trajectories and incorporating natural language reflection, while Guo et al. (2024) prompt LLMs to execute a fixed mutation and crossover process to produce offspring. Fernando et al. (2023) use an LLM to iteratively and self-referentially mutate prompts while improving the mutation prompts themselves. SkillOpt (Yang et al., 2026) instead treats a single agent skill document as trainable external state, using bounded text edits and a held-out validation gate to accept only improving updates. Unlike this single-artifact optimization trajectory, ECS searches populations of contexts assembled directly from a source corpus. Our method differs from existing evolutionary methods in terms of both method and scope. Regarding method, we depart from LLMs as mutation and crossover operators (Guo et al., 2024; Fernando et al., 2023), which are ineffective when the model lacks the knowledge required to meaningfully manipulate task-related information. Instead, we perform mutation with probabilistic draws from the resource pool and crossover with shuffled concatenation, which broadens the exploration space. Regarding scope, we demonstrate evolution's capability as a search method to accumulate knowledge from diverse sources in unseen domains. Where prior work sought to utilize LLMs to iteratively rephrase task queries – adding instructions to show working, answer in full sentences etc – we consider the problem setting of evolving contexts from provided corpora for conferring the knowledge necessary to perform *new skills*. Thus, our work studies evolutionary search over reusable context artifacts rather than only over task prompts or model-generated outputs.

# 3 Evolutionary context search

We present ECS, a framework that evolves raw text resources into high-utility context for LLM skill acquisition. This section formalizes the context-evolution problem and details our pipeline, from the initial construction of context units to the specific GA-style operators used to optimize them for task performance.

## 3.1 Problem Formulation

We formalize context optimization as a search problem. Given text resources $\mathcal{D}$, a target language model $\mathcal{M}$ and a development task set $\mathcal{T}$, we seek an optimal context $C^*$ that maximizes task performance of $\mathcal{M}$.

$$f(C; M, \mathcal{T}) = \mathbf{E}_{(x,y)\in\mathcal{T}}[\mathcal{L}(\mathcal{M}(x, C), y)] \tag{1}$$

where $\mathcal{L}$ is a task-specific scoring metric and $g$ is a function constructing context units from the pool of text resources $\mathcal{D}$, each of which represents an atomic piece of knowledge that can be independently selected, combined, and refined. We search for $C^* = \arg\max_{C \subseteq \mathcal{U}} f(C; \mathcal{M}, \mathcal{T})$, where $\mathcal{U} = g(\mathcal{D})$ is the set of all possible context units. A candidate context $C$ is then a structured combination of these units. ECS therefore searches over $\mathcal{U}$, whereas an authoring method can compress source text into new units.

## 3.2 Algorithm Overview

ECS adapts to diverse tasks by constructing context units in multiple forms (i.e., $g(\mathcal{D})$): raw text from documentation, insights derived from long trajectories, or reusable agentic skills. Prior evolution-based approaches mutate model-generated content, for example rephrasing self-generated instructions, which inherently limits the search space to knowledge the model can already produce. By contrast, ECS draws mutations from provided external text resources, enabling the model to acquire genuinely missing information from its parameters.

---

**Algorithm 1:** Evolutionary Context Search

**Require:** Text resources $\mathcal{D}$, dev set $\mathcal{T}$, target model $\mathcal{M}$, population size $N$, generations $T$

**Ensure:** Optimal context $C^*$

1: $\mathcal{U} \leftarrow g(\mathcal{D})$ {raw text, insights, skills}
2: $P_0 \leftarrow \text{INITIALIZE}(\mathcal{U}, N)$
3: **for** $t = 0$ **to** $T - 1$ **do**
4:     **for all** $C \in P_t$ **do**
5:         $s(C) \leftarrow \text{EVALUATE}(C, \mathcal{M}, \mathcal{T})$
6:     **end for**
7:     $P_{\text{elite}} \leftarrow \text{SELECTELITE}(P_t, s)$
8:     $P_{t+1} \leftarrow \emptyset$
9:     **for** $j = 1$ **to** $N$ **do**
10:         $C_a, C_b \leftarrow \text{SAMPLE}(P_{\text{elite}}, s)$
11:         $C' \leftarrow \text{CROSSOVER}(C_a, C_b)$
12:         $C' \leftarrow \text{MUTATE}(C', \mathcal{U})$
13:         $C' \leftarrow \text{REFINE}(C', \mathcal{T})$ {LLM-guided}
14:         $P_{t+1} \leftarrow P_{t+1} \cup \{C'\}$
15:     **end for**
16: **end for**
17: **return** $C^* \leftarrow \arg\max_{C \in P_T} \text{EVALUATE}(C, \mathcal{M}, \mathcal{T})$

---

Algorithm 1 presents the complete ECS procedure. The algorithm operates in two phases: initialization and evolution. During initialization, we construct the initial pool $\mathcal{U}$ from text resources $\mathcal{D}$, extracting units at varying abstraction levels depending on the task setting — from verbatim source texts to distilled insights and reusable skills (Sec 3.3). We then sample an initial population $P_0$ of $N$ candidate contexts by drawing units from $\mathcal{U}$ without replacement, ensuring broad coverage of the knowledge pool while maintaining comparable context lengths across candidates.

The evolution loop iteratively improves the population $P_t$. Each generation evaluates all candidates on the development set $\mathcal{T}$ and selects top performers as elite contexts $P_{\text{elite}}$. We sample parents fitness-proportionally from $P_{\text{elite}}$ and produce offspring through crossover. Mutation then introduces variation by adding or replacing units to explore the full unit space $\mathcal{U}$. Mutation and crossover thereby jointly expand the population of performant contexts, as determined by those contexts contribution to advancing $\mathcal{M}$ performance on the task. Finally, LLM-guided refinement resolves logical contradictions inside each offspring. After $T$ generations, we return the highest-performing context $C^*$.

### 3.3 Context Unit Construction

Different tasks require knowledge at different granularities: code generation benefits from exact syntax and precise documentation, while role-play benefits from abstracted principles. We therefore design context units at varying abstraction levels, allowing ECS to evolve units that span these differing levels of abstraction as necessitated by different tasks. We define three representative types below.

**Source texts.** These units preserve the precise syntax from source materials, which is vital for replicating exact phrasing or code patterns. For example, in Domain-Specific Language tasks, we include code files from NVIDIA's CuTe tutorials, maintaining exact API usage patterns that the model must reproduce.

**Insights.** These units distill abstract principles from the provided text source. Typically, these are actionable rules that capture patterns the model should follow or pitfalls to avoid. For example, given failed trajectories from $\tau^2$-bench (Barres et al., 2025), we prompt another model to analyze the errors and extract rules such as "when processing refunds, retrieve the payment directly from the history rather than prompting the user."

**Skills.** These units encode reusable procedural knowledge as modular, callable actions. Each skill packages domain-specific instructions and multi-step workflows that the model can invoke on demand. We adapt the Agent Skills format (Anthropic, 2025b), which provides a structured representation for packaging procedural knowledge. An example is "write-mha-cutedsl-kernel," which encapsulates the procedure for writing multi-head attention kernels in CuTe DSL. While skills offer a natural structure for procedural knowledge, naively including all available skills can degrade performance due to context distraction. ECS partially mitigates

this by automatically curating task-relevant skills (see Section 4), though improving skill representation and invocation remains an open direction.

## 3.4 Evolutionary Operators

We now detail each evolutionary operator which jointly comprise our context search method.

**Initialization.** We initialize population $P_0$ of $N$ candidates (typically 32) by sampling units from $\mathcal{U}$ uniformly without replacement. Each candidate context starts with a predetermined number of units, which we set based on task characteristics: tasks with long units (e.g., complete code files) use fewer units per context, while tasks with short units (e.g., concise insights) use more. This maintains comparable context lengths across candidates while preserving diversity among individuals in the initial population.

**Fitness Evaluation.** We score each context $C$ on the development set $\mathcal{T}$ by querying the LLM $\mathcal{M}$ with $C$ as context across all tasks in the development set $\mathcal{T}$. The fitness $s(C)$ equals the task success rate, normalized to $[0, 1]$. We use a single rollout per task, which we find sufficient in practice, though additional rollouts could further reduce variance from stochastic $\mathcal{M}$ outputs.

**Selection.** Selection pressure drives the population toward higher-performing contexts. We first select the top fraction (e.g., 60%) of contexts as the elite set $P_{\text{elite}}$. We then repeatedly sample parent pairs from this elite set using fitness-proportional selection until we generate $N$ offspring for the next generation. For any context $C$, its selection probability $p_C$ equals its fitness normalized by the elite set's total fitness, $p_C = \frac{s(C)}{\sum_{K \in P_{elite}} s(K)}$.
This focuses reproduction on top performers while maintaining variation within elites.

**Crossover.** Crossover combines units from two parent contexts $C_a$ and $C_b$ to produce one offspring. We concatenate all units from both parents; if the combined set exceeds the maximum context size, we randomly sample from the concatenated context unit pool. This allows offspring to inherit complementary information from both parents while respecting context length constraints.

**Mutation.** Mutation introduces variation controlled by a per-context mutation rate (default is 0.1). When mutation triggers, we sample a new unit from the full pool $\mathcal{U}$. If the context has not reached its maximum size, we add the new unit; otherwise, we replace a randomly selected existing unit. This operator ensures the algorithm explores the entire unit space and can escape local optima.

## 3.5 LLM-Guided Refinement

After mutation, an LLM reviews each offspring context to identify and resolve logical contradictions. The LLM receives the offspring context and instructions to detect inconsistencies; in principle, failed task examples could also be provided to guide refinement, though we omit this for simplicity in our experiments. This refinement step addresses a limitation of blind recombination: merged units may contain conflicting guidance. The LLM identifies contradictions and either removes or resolves the conflicts.

## 3.6 Convergence Analysis

To explain the rapid convergence observed in our experiments, we analyze an idealized variant of ECS that captures its core search dynamics. The analysis abstracts away population-level crossover and LLM-guided refinement, and focuses on the core mechanism shared with our method: fitness-guided selection over fixed-size context sets with random replacement mutation. Let $\mathcal{U}$ be a pool of $N$ context units, let $T^\star \subseteq \mathcal{U}$ denote an optimal target set of size $r$, and let each candidate context $S_t \subseteq \mathcal{U}$ have capacity $k$, where $r \leq k < N$. Define $X_t = |S_t \cap T^\star|$ as the overlap between the current context and the target set. Under a monotone utility assumption in which adding a target unit strictly improves expected task performance beyond bounded evaluation noise, elitist selection makes $\{X_t\}$ non-decreasing. Thus, the search process is a pure birth Markov process.

**Proposition 3.1** (Log-linear convergence). *Assume fixed-size random replacement mutation and elitist selection, with excess capacity $k \geq (1 + \gamma)r$ for some constant $\gamma > 0$. If the utility has the form $U(S) = g(|S \cap T^\star|) + b(S)$, where $|b(S)| \leq \epsilon$ and $g(i + 1) - g(i) \geq \beta > 2\epsilon$ for all $i \in \{0, \ldots, r - 1\}$, then the expected*

*time to recover all $r$ target units satisfies*

$$\mathbb{E}[\tau_r] = \mathcal{O}(N \log r).$$

The result connects ECS to classical analyses of elitist evolutionary algorithms and absorbing Markov chains (Rudolph, 1994; Droste et al., 2002), which establish logarithmic-factor convergence rates for monotone pseudo-Boolean optimization. In our setting, the proposition suggests that when useful context units provide reliably positive marginal utility, fitness-guided context search can identify high-utility combinations in few generations. We provide the full proof and discussion in Appendix F.

## 4 Experiments

We empirically validate the benefits of ECS in skill acquisition. We evaluate our method on coding in unseen domain-specific languages with BackendBench (Saroufim et al., 2025), on multi-turn agentic user assistance with $\tau^2$-Bench (Barres et al., 2025), and on terminal tasks with Terminal-Bench (Merrill et al., 2026), using Gemini-3-Flash (Gemini Team, 2025a) as the base model for context search. We also assess cross-model transfer by applying the discovered contexts to Claude-4.5-Sonnet (Anthropic, 2025a) and DeepSeek-V3.2 (Liu et al., 2025). In addition, Appendix H shows that the SFT configurations in our sweep were unstable and did not yield performance improvements in our data-scarce setting.

### 4.1 Experimental Setup

**Baselines.** We evaluate ECS against several baselines. We implement RAG baselines with LlamaIndex (Liu, 2022), varying two primary axes: chunking strategy and retrieval method. Chunking strategies include fixed-size splitting (**Chunk**; 1,024 tokens with 200-token overlap) and **AST**-based parsing, which segments code at semantic boundaries. Retrieval methods include **Dense** (similarity over OpenAI text-embedding-3-small embeddings (OpenAI, 2024)), **BM25** (sparse keyword matching), and **Hybrid** (reciprocal rank fusion of Dense and BM25). We sweep $k \in \{5, 10, 20\}$ for Chunk + Dense and find $k = 10$ performs best, which we use for all RAG configurations. We also include **Full Context**, which loads all available documentation into the context window, and **Random Sample**, which randomly selects the same number of context units as ECS.

**Tasks.** We evaluate our method on three challenging benchmarks: kernel coding in a new Domain-Specific Language (DSL), multi-turn agentic user assistance, and heterogeneous tasks executed in containerized terminal environments.

*BackendBench (CuTeDSL)* (Saroufim et al., 2025) focuses on testing whether models can generate correct GPU kernels in various DSLs (Triton, CUDA, CuTeDSL). Among these, we choose CuTeDSL to showcase ECS capacity to convert newly released textual tutorials into knowledge-rich context capable of guiding the model. CuTeDSL is NVIDIA's Python DSL built on CUDA Templates for Linear Algebra Subroutines (CUTLASS) (NVIDIA, 2026), where text resources are provided by tutorial examples in the repository. We randomly select 20 core PyTorch operators from PyTorch's operator information (OpInfo) suite; after suite and dtype filtering, the number of retained correctness inputs varies by operator. Appendix C.1 details these operators. We run 3 evaluations and report the average correctness rate.

*$\tau^2$-Bench (Airline Domain)* (Barres et al., 2025) evaluates agents on completing user requests through multi-turn interaction, while adhering to domain-specific policies which may conflict with user requests. The airline domain presents customer service tasks – booking modifications, cancellations, and policy inquiries – that must be completed while maintaining airline policy standards. We obtain text resources by collecting trajectories from GPT-5.2 (OpenAI, 2025) and Gemini-3-Pro (Gemini Team, 2025b) on the official training set, then prompting Gemini-3-Flash to extract insights from these trajectories. Following Barres et al. (2025), we use GPT-4.1-2025-04-14 (Achiam et al., 2023) as the user simulator. Pass$^k$ for $k \in \{1, 2, 3\}$ measures the rate at which all $k$ trials succeed. We evaluate each configuration 3 times with 3 trials, yielding 9 runs total.

*Terminal-Bench 2.1* (Merrill et al., 2026) evaluates agents on realistic tasks in containerized terminal environments, with task-specific tests providing success signals. We construct a fixed category and difficulty stratified split of 53 training and 36 held-out test tasks. Following the same trajectory-to-insight loop used

for $\tau^2$-Bench, we collect one Gemini-3-Flash baseline trajectory and extract one insight per training task. We evaluate with the Terminus-2 agent and report mean held-out accuracy over three runs.

Appendix B provides the full ECS configuration. Train/dev/test splits are kept strictly disjoint; split documentation is in Appendix A, significance tests for BackendBench and $\tau^2$-Bench in Appendix C.4, development-set size sensitivity in Appendix C.5, and hyperparameter sensitivity in Appendix E.2.

### 4.2 Main Results

**Observation 1: Evolutionary Context Search constructs more effective context than retrieval-based approaches.** ECS consistently outperforms standard retrieval baselines across the DSL kernel coding and agentic tasks.

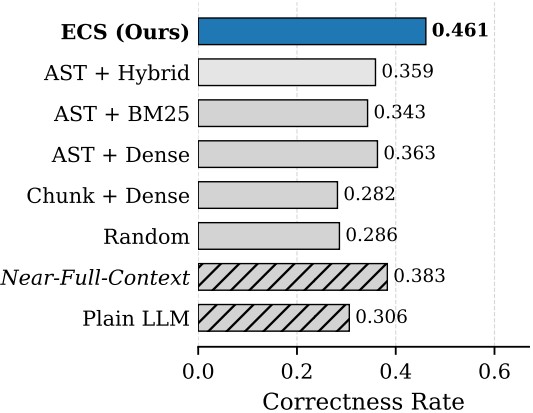 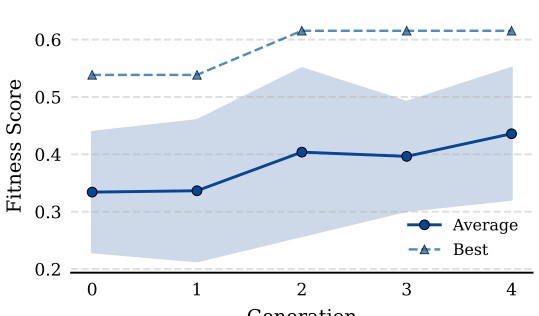

Figure 2: **Left:** Performance comparison on BackendBench. Hatched bars denote baselines with distinct context budgets. Near-Full-Context processes 1,003.5k context tokens, whereas Plain LLM uses no extra tokens. All other methods, including ECS, are limited to at most 10 retrieved documents, corresponding to approximately 50k context tokens. ECS achieves a 27% relative improvement over AST+Dense. Detailed results in Appendix C.2. **Right:** Fitness score during the evolutionary search process.

Regarding BackendBench, where the model must master CuTeDSL using tutorial-based coding resources, Figure 2 (left) shows that ECS achieves a correctness rate of 0.461, outperforming the strongest retrieval baseline (AST + Dense) by a 27% relative improvement. Relative to AST + Dense, ECS improves all three operator-family averages: from 0.759 to 0.857 for trigonometric operators, from 0.158 to 0.231 for arithmetic, and from 0.069 to 0.180 for linear algebra. Appendix C.3 provides the complete breakdown and a case analysis of BMM. We also illustrate the training dynamics in Figure 2 (right). The rapid improvement in the fitness curve is consistent with the convergence behavior predicted by Proposition 3.1 under monotone marginal utility assumptions. Beyond this, several patterns emerge. First, the chunking strategy is critical: all AST-based methods consistently outperform Chunk + Dense, confirming that preserving semantic boundaries matters for code documentation. Second, retrieval method choice has limited impact—Dense, Hybrid, and BM25 perform similarly when paired with AST. Third, naive approaches can hurt: both Chunk + Dense and Random underperform the Plain LLM baseline. Since Random uses the same context budget as ECS, this confirms that arbitrary selection introduces noise, which degrades performance. This concurs with existing findings that irrelevant context deteriorates RAG's performance relative to the base model (Yoran et al., 2023), implying that RAG is dependent on careful curation. These results expose a core limitation of retrieval: optimizing chunk-level relevance rather than compositional coherence. ECS addresses this by evolving context combinations that maximize task performance.

On the agentic $\tau^2$-*Bench* in Table 1, ECS achieves 0.756 on Pass[1], compared with 0.717 for the strongest baseline, Full Context. Two observations stand out. First, the gap widens under stricter metrics: on Pass[3], ECS maintains 0.667 while Plain LLM and BM25 degrade sharply. This indicates that curated contexts enable more consistent policy adherence across repeated trials. Second, Full Context outperforms the retrieval-based

methods, suggesting that broader coverage matters for this policy-heavy task; ECS is higher than Full Context in this setting while using far less context. This reveals a further benefit of ECS, which is that our method constructs efficient contexts that confer the required knowledge without context bloat. Terminal-Bench broadens this main result to heterogeneous terminal tasks (Table 1). ECS reaches 0.611 accuracy, compared with 0.602 for Full Context, 0.556 for Random, 0.537 for BM25 RAG, and 0.519 for the Plain LLM. Full Context underperforms ECS, while using all 53 units rather than at most ten.

Table 1: **Main agentic benchmark results.** ECS outperforms the evaluated baselines on $\tau^2$-Bench and Terminal-Bench 2.1. $\tau^2$-Bench values are means over three independent evaluations; Welch tests are reported in Appendix 11. Terminal-Bench values are mean held-out accuracy over three runs.

| Method | $\tau^2$-Bench | | | Terminal-Bench 2.1 |
|---|---|---|---|---|
| | **Pass$^1$** ↑ | **Pass$^2$** ↑ | **Pass$^3$** ↑ | **Accuracy** ↑ |
| Plain LLM | 0.622 | 0.489 | 0.400 | 0.519 |
| Random | 0.650 | 0.556 | 0.500 | 0.556 |
| BM25 | 0.641 | 0.550 | 0.483 | 0.537 |
| Full | 0.717 | 0.628 | 0.550 | 0.602 |
| **ECS (Ours)** | **0.756** | **0.700** | **0.667** | **0.611** |

**Observation 2: Evolved contexts can retain value across held-out models, with model- and benchmark-dependent gains.** To assess the generalizability of ECS beyond the base model, we evaluate contexts evolved with Gemini-3-Flash on two powerful held-out models: Claude-4.5-Sonnet and DeepSeek-V3.2.

On *BackendBench* (Figure 3), the evolved context reaches 0.611 correctness with Claude-4.5-Sonnet, compared with 0.564 for AST+Dense. With DeepSeek-V3.2, it reaches 0.223, compared with 0.031 for Plain and 0.065 for AST+Dense. We provide further analysis on this context transfer in Section 5.2.

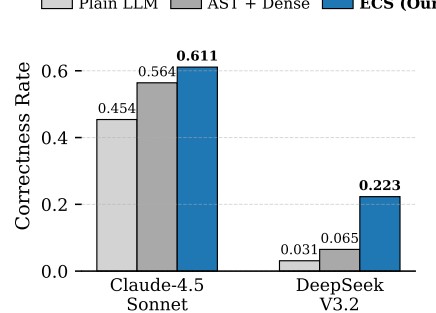

Figure 3: **Transferability to held-out models (BackendBench).** Contexts evolved with Gemini-3-Flash improve performance when supplied to hold-out models.

The $\tau^2$-Bench transfer results demonstrate substantive gains for Claude-4.5-Sonnet, while revealing limiting cases of transfer in the case for DeepSeek-V3.2 (Table 2). With Claude-4.5-Sonnet, ECS matches Full on Pass$^1$ and is higher on Pass$^3$ (0.583 vs. 0.500). With DeepSeek-V3.2, ECS reaches Pass$^1$ comparable to Full (0.600 vs. 0.633) using 6× fewer tokens (848 vs. 5491), and matches Plain on Pass$^2$/Pass$^3$. Taken together, these results demonstrate that ECS captures transferable contexts enabling skill acquisition—even when evolved with a smaller model and then transferred to more capable models. In *BackendBench*, success requires precise knowledge of code structure and protocols, which are likely to be beneficial regardless of model capability or size. Hence we observe transfer gains across both DeepSeek and Claude. Conversely, $\tau^2$-*Bench* relies on discursive styles and adherence to less concrete policy guidelines, for which it is less obvious that the contexts evolved via a smaller, less capable model may necessarily transfer to larger, more capable models. In this sense, $\tau^2$-*Bench* probes the limiting case of transfer, where we find, encouragingly, that even in this setting ECS enables Claude to access substantial performance and DeepSeek to attain similar performance to *Full* at one sixth of the token usage.

**Observation 3: Evolutionary Context Search enables effective skill-based augmentation by automatically curating task-relevant skills.** Agentic skills represent an emerging paradigm for extending LLM capabilities (e.g., Claude Skills), but best practices for skill selection remain an open challenge. To evaluate ECS in this setting, we extract 75 skills from the CuTeDSL documentation, each corresponding to an example code snippet. From our experiments, we find that naively including all available skills degrades performance (0.283) compared to the plain LLM baseline (0.306), due to context distraction caused by

Table 2: **Transferability on $\tau^2$-Bench.** With Claude-4.5-Sonnet, ECS matches the Full baseline on Pass[1], outperforms it on Pass[2], and significantly outperforms it on Pass[3] ($p = 0.025$). Notably, ECS achieves comparable performance to the Full baselines with fewer tokens, reducing usage from 5491 to 848.

| Metric | Claude-4.5-Sonnet | | | DeepSeek-V3.2 | | |
|---|---|---|---|---|---|---|
| | **Plain** | **Full** | **ECS** | **Plain** | **Full** | **ECS** |
| Pass[1] | 0.600 | 0.678 | **0.678** | 0.583 | **0.633** | 0.600 |
| Pass[2] | 0.506 | 0.561 | **0.617** | 0.478 | **0.500** | 0.472 |
| Pass[3] | 0.450 | 0.500 | **0.583** | 0.417 | 0.417 | 0.417 |
| # Tokens | 0 | 5491 | 848 | 0 | 5491 | 848 |

irrelevant snippets. By contrast, ECS successfully filters this noise, achieving the highest correctness rate (0.310), demonstrating that ECS can serve as a complementary technique for skill-based systems—transforming noisy skill pools into effective context through evolutionary curation.

## 4.3 Beyond Static Retrieval

The baselines compared above all share a fixed-pipeline structure: a similarity-based retriever returns chunks for the query. To check that ECS's lead is not an artifact of comparing only against static retrieval, we evaluate against four paradigms that move past this design in different directions—dynamic retrieval (FLARE), LLM-based prompt optimization (OPRO), controlled skill optimization (SkillOpt), and direct LLM context curation. Because each makes different setup demands (e.g., FLARE requires output-token log-probabilities), we evaluate each on the benchmark where it is best instrumented and treat the four as independent probes rather than a unified leaderboard. FLARE retrieves adaptively during generation; OPRO uses an LLM to iteratively propose and select prompts; Direct LLM curation asks the model to assemble a static context from the supplied pool; SkillOpt applies validation-gated edits to a single skill document. Appendix D provides the complete protocols, constraints, and optimization budgets.

**Dynamic retrieval (FLARE on BackendBench).** Because FLARE (Jiang et al., 2023) requires output-token log-probabilities unavailable from Gemini-3-Flash, we compare all methods in this experiment using Qwen3.5-122B-A10B (Yang et al., 2025). As Table 3 shows, FLARE (0.198) improves over Plain (0.063) but falls slightly behind standard dense RAG (0.227): BackendBench's official evaluation wraps each operator request in a verbose system+user prompt that dominates the retrieval signal regardless of when retrieval is triggered. ECS (0.278) preserves its lead by sidestepping query-conditional retrieval entirely.

Table 3: **Open-source comparison on BackendBench with Qwen3.5-122B-A10B.** FLARE improves over Plain but falls behind standard dense RAG due to verbose, multi-part queries. ECS continues to lead by a substantial margin.

| Method | Correctness Rate |
|---|---|
| Plain LLM | 0.063 |
| FLARE | 0.198 |
| AST + Dense RAG | 0.227 |
| **ECS (Ours)** | **0.278** |

**Additional optimization baselines on $\tau^2$-Bench.** We evaluate three additional baselines that isolate alternative explanations for ECS's gains. With the same insight pool and a matched compute budget, OPRO (Yang et al., 2024) improves over Plain LLM but trails ECS at every Pass level, with the gap widening under stricter evaluation (Pass[3]: 0.500 vs. 0.667; Table 4). Controlled skill optimization with SkillOpt (Yang et al., 2026), using the same 30-task development pool and freezing the learned skill before testing, similarly falls behind ECS by 0.089, 0.133, and 0.167 on Pass[1], Pass[2], and Pass[3], respectively. Finally, direct Gemini-3-Flash context curation with matched corpus access and iteration budget improves

over Plain LLM (Pass[1]: 0.678 vs. 0.622) but remains below ECS at every Pass level. Together, these results suggest that ECS's gains arise not merely from rephrasing a fixed instruction set or providing an LLM with extended corpus access, but from iteratively selecting and integrating knowledge across heterogeneous policies. Appendix D.1 provides the full SkillOpt setup and realized rollout cost.

Table 4: **Adaptive baselines on $\tau^2$-Bench.** OPRO, direct LLM context curation, and SkillOpt improve over Plain but fall short of ECS. Values are means over three independent evaluations.

| Method | Pass[1] | Pass[2] | Pass[3] |
|---|---|---|---|
| Plain LLM | 0.622 | 0.489 | 0.400 |
| OPRO | 0.694 | 0.578 | 0.500 |
| LLM Curation (Flash) | 0.678 | 0.578 | 0.533 |
| SkillOpt (Flash) | 0.667 | 0.567 | 0.500 |
| **ECS (Ours)** | **0.756** | **0.700** | **0.667** |

Across these four comparisons, ECS outperforms the evaluated dynamic-retrieval, prompt-optimization, direct-curation, and skill-optimization baselines. These results support population-based context search in the settings tested, but do not establish that evolutionary search is universally preferable to agentic skill induction or corpus exploration. Mechanisms may also be complementary: authoring methods can synthesize compact candidate units, while ECS can select and compose these units using the performance of downstream tasks.

## 5 Analysis and Ablation

In this section, we examine the properties of ECS. We first analyze the evolved context qualitatively, then study how models utilize contexts. Finally, we conduct ablation studies and analyze computational costs.

### 5.1 Analysis of evolved context.

Figure 4 presents the context discovered with Gemini-3-Flash from the CuTeDSL code tutorial. ECS selected 7 spanning kernel implementations, GEMM/MHA examples, and interface code. At the kernel

**(a) hopper/fmha.py** `[Kernel]`                                       **2,540 lines (58%)**

*Role: Fused MHA with TMA + TensorCore*

```
1  # Q*K^T, softmax, softmax(Q*K^T)*V fused
2  for j in cutlass.range_constexpr(
3      cute.size(acc_qk_mn, mode=[1])):
4      acc_qk_mn[i, j] = cute.math.exp2(
5          scale_softmax_log2 * acc_qk_mn[i, j] - scale_max,
6          fastmath=True)
```

**(b) tensorop__gemm.py** `[Kernel]`                                   **1,012 lines (23%)**

*Role: Dense GEMM for Ampere architecture*

```
1  # Creates MMA atom with 16x8x16 shape for MNK
2  op = cute.nvgpu.warp.MmaF16BF16Op(
3      self.ab_dtype, self.acc_dtype, self.mma_inst_shape)
4  tC = cute.make_layout(self.atom_layout_mnk)
5  tiled_mma = cute.make_tiled_mma(op, tC, permutation_mnk)
```

**(c) jit__argument.py** `[FFI]`                                           **320 lines (7%)**

*Role: C-struct tensor interface via LLVM*

```
1  # FFI: Extract pointer from C-struct
2  ptr_val = llvm.extractvalue(
3      llvm.PointerType.get(), self, [0], loc=loc, ip=ip)
4  return cute.make_ptr(cutlass.Float32, ptr_val)
```

Figure 4: **Core components of the BackendBench evolved context.** The three largest code snippets in the selected context are: **(a)** a fused MHA kernel targeting the NVIDIA Hopper architecture (58% of context); **(b)** a dense GEMM kernel configuration for Ampere Tensor Cores (23%); and **(c)** a low-level FFI interface managing LLVM pointer extraction (7%).

layer, `hopper/fmha.py` provides a comprehensive example of a fused mega-kernel—contributing TMA-based

memory transfers, warp-specialized execution, and tensor core MMA(Matrix Multiply-Accumulate) patterns, while also demonstrating the `cute.math.*` API for arithmetic operations. The second kernel example, `tensorop_gemm.py`, shows how to construct tiled MMA operations, covering atom layout configuration and threadblock rasterization. Finally, at the FFI (Foreign Function Interface layer), `jit_argument.py` defines C-struct tensor interfaces, enabling data passing between Python and compiled kernels.

## 5.2 Context utilization across models.

| (a) Evolved Context | (b) RAG Baseline | (c) With Evolved Context |
|---|---|---|
| *hopper/fmha.py (lines 1372–1384)* | *DeepSeek-V3.2 generates wrong API* | *DeepSeek-V3.2 generalizes correctly* |

```
1  for j in range_constexpr(...):
2    acc_qk_mn[i,j] = cute.math.exp2(
3      scale_softmax_log2
4      * acc_qk_mn[i,j]
```
`cute.math.*` pattern

```
1  input_val = gA[tidx]
2  # WRONG: ArithValue is symbolic
3  result = math.atan(input_val)
```
✗ Fails: stdlib on symbolic value

```
1  input_val = gA[tidx]
2  # CORRECT: DSL intrinsic
3  result = cute.math.atan(input_val)
```
✓ Works: DSL intrinsic

| Operator | RAG Baseline | Evolved Context | Knowledge Transferred |
|---|---|---|---|
| `atan.default` | 0% (0/1) | **100%** (1/1) | `cute.math.atan()` |
| `atan2.default` | 0% (0/9) | **88.9%** (8/9) | `cute.math.atan2()` |
| `div.Tensor` | 0% (0/18) | **88.9%** (16/18) | `cute.math.*` pattern |
| **Overall (20 ops)** | 6.5% | **22.3%** | 3.4× improvement |

Figure 5: **Cross-model context transfer from Gemini-3-Flash to DeepSeek-V3.2 on BackendBench.** **Top:** (a) Evolved context demonstrates the `cute.math.*` pattern. (b) Without this context, DeepSeek-V3.2 incorrectly uses Python's `math.atan()`, causing JIT compilation failures. (c) With evolved context, DeepSeek-V3.2 correctly generalizes to DSL intrinsics. **Bottom:** Quantitative results show that the evolved context improves pass rate from 6.5% to 22.3% (3.4×); e.g., `div.Tensor` passes 16 out of 18 test cases.

We analyze how exactly the evolved context enables knowledge transfer across models. Figure 5 illustrates one specific example for BackendBench. The evolved context contains `fmha.py`, which has the `cute.math.exp2()` function for softmax operation. Although this example never mentions `atan` or other trigonometric functions, it implicitly teaches the correct API pattern: math operations in CuTeDSL require DSL intrinsics under `cute.math.*`, not Python's standard library. When DeepSeek-V3.2 receives this context, it extracts this structural pattern and generalizes it to unseen operators, correctly generating `cute.math.atan()` despite never observing this specific function in the provided examples.

In contrast, RAG retrieval for the `atan` operator returns semantically relevant but functionally useless context: GEMM kernels, elementwise addition, and tensor utilities—none of which contain any `cute.math.*` calls. Without working examples, DeepSeek falls back to Python's `math.atan()`. This fails because inside a `@cute.kernel`, tensor elements are symbolic `ArithValue` objects, not concrete Python values. Only DSL intrinsics, such as `cute.math.atan()`, are recognized by the JIT and translated to CUDA device code.

## 5.3 Ablation Study

To understand the contribution of each component, we evaluate ECS variants with fitness-guided selection, mutation, and refinement individually removed. Figure 6 shows the results. Fitness-guided selection is critical: removing it causes the largest degradation on both benchmarks, with BackendBench dropping to 0.286 (below Plain LLM). Mutation also proves essential, reducing performance on both tasks when removed. Interestingly, refinement exhibits domain-dependent behavior: on BackendBench, its removal has negligible impact (0.461 vs 0.458), whereas on $\tau^2$-Bench it causes substantial degradation (Pass[3]: 0.667 vs 0.488). This aligns with our earlier analysis, LLMs struggle to filter tutorial coding samples and tend to retain all of them, while insights often contain explicit contradictions (e.g., conflicting refund rules) that LLM can readily identify.

## 5.4 Computational cost.

**Search Cost.** ECS incurs an initial computational cost to discover the optimal context $C^*$. In our experiments, the search budget was limited to $T = 10$ generations with a population size of $N = 32$, resulting

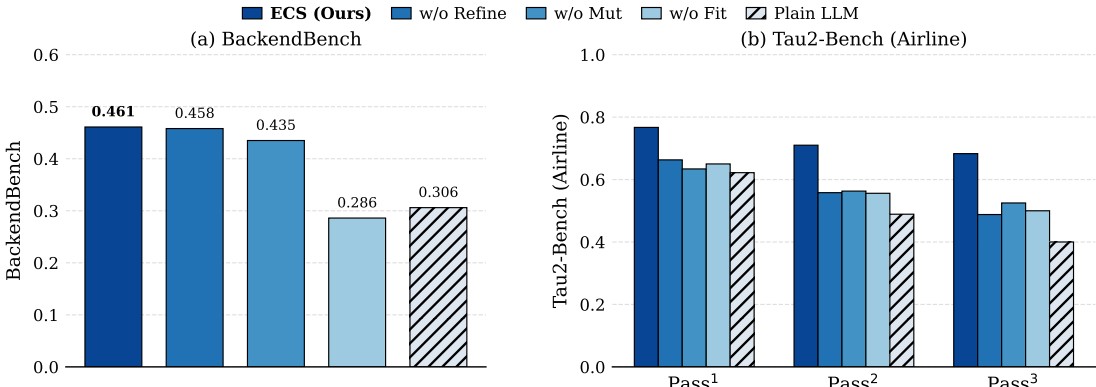

Figure 6: **Ablation study.** We evaluate variants by removing components. *Fitness* and *Mutation* are universally critical. *Refinement* is domain-dependent: essential for resolving conflicting agentic policies ($\tau^2$-Bench) but with negligible impact on BackendBench.

in approximately 320 evaluations on the development set (fewer on BackendBench, which uses $T = 5$). While this exceeds the setup cost of standard RAG, which requires computing embeddings for dense retrieval, it is orders of magnitude cheaper than typical SFT or RL workflows, which often necessitate tens of thousands of rollouts to stabilize policy updates Shao et al. (2024). Crucially, ECS requires only black-box inference calls, avoiding the memory overhead of gradient updates and optimizer states required by gradient-based methods. Relative to LLM-based prompt optimizers such as OPRO, ECS uses rule-based evolutionary operators and an optional deletion-only refiner, eliminating candidate-generation model calls when refinement is disabled and reducing its output tokens by over 80% when it is enabled; full analysis are provided in Appendix E.3. **Inference Efficiency.** Once the optimal context is discovered, ECS offers efficiency gains over retrieval baselines during deployment. Because the evolved context $C^*$ remains static across all incoming queries for a given task, it is fully compatible with Context Caching (KV-Caching). In contrast, RAG systems retrieve different context chunks for every unique query, preventing the model from hitting the cached prompt prefix. Current industry pricing for cached context (e.g. Claude 4.5) charges approximately 10% of the standard input token price for cache hits Anthropic (2024). Furthermore, caching eliminates the prefill calculation for the context, reducing Time-To-First-Token. Thus, while ECS requires an upfront search investment, it reduces the recurring deployment cost for the context portion compared to dynamic retrieval methods.

## 6 Conclusions

**Summary.** We introduce Evolutionary Context Search (ECS), a method that reframes knowledge acquisition as an evolutionary search over text resources. ECS improves BackendBench macro-average correctness by 27% relative to the strongest RAG baseline and surpasses the non-evolutionary baseline on $\tau^2$-Bench and Terminal Bench. Relative to Plain LLM, the BackendBench improvement is concentrated in trigonometric operators. Contexts evolved using Gemini-3-Flash also retain value on held-out models, although the transfer magnitude depends on the model and benchmark.

**Limitation.** Several factors bound ECS's applicability. Overall, ECS is most compelling when the domain is stable, the development set represents deployment, a reliable automatic task metric is available as fitness, and the resulting context will be reused enough to amortize search; rapidly changing or one-off settings may instead favor dynamic retrieval or direct authoring. First, the development set must be representative of the deployment distribution, since task accuracy on it drives fitness; non-representative dev sets risk overfitting the evolved context. Second, the transfer guarantee rests on the aligned-marginal-utility condition (Appendix G), which can weaken when source and target models benefit from disjoint structural patterns. Third, the upfront search cost amortizes only when the evolved context is reused across many queries; in single-query regimes, lightweight retrieval may still be preferable.

**Future Work.** The transfer results motivate future work on using ECS as one component of an automated data-curation pipeline for SFT. Such work should compare evolutionary selection directly with agentic authoring and test whether SFT can internalize the selected demonstrations.

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

## A  Data Splits

We keep the data used to drive evolutionary search strictly disjoint from the data used to report final scores in all three benchmarks.

$\tau^2$**-Bench (airline).**  We use the airline domain, which provides two official splits: a 30-task train split and a 20-task test split. The 30 train tasks are used solely for fitness evaluation during search and for constructing insight units (including the trajectory pool used by another model to distill rules). The 20 test tasks are used *exclusively* for the numbers reported in the main paper; no test task is ever shown to ECS during search.

**BackendBench (CuTeDSL).**  The 20 PyTorch operators in our evaluation suite (Appendix C.1) are used exclusively for reporting. For fitness evaluation during ECS search we use 10 development samples drawn from a disjoint development pool of operators not included in the evaluation suite; no test case from the 20 reporting operators is ever shown to ECS during search.

**Terminal-Bench 2.1.**  We partition its 89 tasks into a fixed 53-task training split and a disjoint 36-task test split, stratified by category and difficulty with seed 300. Only training-task trajectories are used to construct the 53-unit insight pool, and only a fixed ten-task subset of the training split is used for search fitness. The 36 test tasks are used exclusively for the final three-run comparison in Table 1.

## B  ECS Configuration

We run ECS for 5 generations (10 for $\tau^2$-Bench) with a population size of 32, selecting the top 60% as elites with a mutation rate of 0.1. Each context is limited to a maximum of 10 units, drawn from a pool of 85 source documents for BackendBench and 60 extracted insights for $\tau^2$-Bench. Fitness is evaluated on 10 development samples for BackendBench and on the 30 official training tasks for $\tau^2$-Bench. For BackendBench and $\tau^2$-Bench, $\mathcal{M}$ is Gemini-3-Flash, and we use Gemini-3-Flash itself for refinement (the REFINE step in Algorithm 1), keeping the entire pipeline within a single model so that any reported gains are not driven by an auxiliary, more capable refiner. For Terminal-Bench, we use five generations, a population of eight, an elite fraction of 0.5, and no LLM refinement; candidates contain at most ten units from its 53-unit pool and are evaluated on a fixed ten-task subset of the training split.

## C   BackendBench Details

### C.1   Evaluation Operators

We evaluate our method on 20 PyTorch operators from the BackendBench benchmark, spanning three categories: trigonometric functions (8), arithmetic operations (4), and linear algebra primitives (8). These operators represent a diverse set of computational patterns commonly encountered in deep learning workloads. Table 5 provides detailed descriptions of each operator.

### C.2   Full BackendBench Results

Table 5: PyTorch operators used in BackendBench evaluation. We evaluate 20 operators across three categories: trigonometric functions (8 ops), arithmetic operations (4 ops), and linear algebra primitives (8 ops).

| Category | Operator | Signature | Description |
|---|---|---|---|
| *Trigonometric Functions* | | | |
| | acos | $f : [-1, 1] \to [0, \pi]$ | Computes element-wise inverse cosine (arccosine) of the input tensor. |
| | acosh | $f : [1, \infty) \to [0, \infty)$ | Computes element-wise inverse hyperbolic cosine: $\ln(x + \sqrt{x^2 - 1})$. |
| | asin | $f : [-1, 1] \to [-\frac{\pi}{2}, \frac{\pi}{2}]$ | Computes element-wise inverse sine (arcsine) of the input tensor. |
| | asinh | $f : \mathbb{R} \to \mathbb{R}$ | Computes element-wise inverse hyperbolic sine: $\ln(x + \sqrt{x^2 + 1})$. |
| | atan | $f : \mathbb{R} \to (-\frac{\pi}{2}, \frac{\pi}{2})$ | Computes element-wise inverse tangent (arctangent) of the input tensor. |
| | atan2 | $f : \mathbb{R}^2 \to [-\pi, \pi]$ | Computes element-wise two-argument arctangent of $y/x$, using signs to determine quadrant. |
| | atanh | $f : (-1, 1) \to \mathbb{R}$ | Computes element-wise inverse hyperbolic tangent: $\frac{1}{2} \ln \frac{1+x}{1-x}$. |
| | ceil | $f : \mathbb{R} \to \mathbb{Z}$ | Rounds each element to the smallest integer greater than or equal to the input. |
| *Arithmetic Operations* | | | |
| | div | $(a, b) \mapsto a/b$ | Computes element-wise division. Supports both integer and floating-point semantics. |
| | div_mode | $(a, b, \text{mode}) \mapsto a/b$ | Element-wise division with explicit rounding: `trunc` or `floor` mode. |
| | fmod | $(a, b) \mapsto a - b \cdot \text{trunc}(a/b)$ | C-style remainder (truncated division). Result has same sign as dividend. |
| | remainder | $(a, b) \mapsto a - b \cdot \text{floor}(a/b)$ | Python-style remainder (floored division). Result has same sign as divisor. |
| *Linear Algebra Primitives* | | | |
| | addmm | $\beta M + \alpha(A@B)$ | Matrix-matrix multiply with accumulation. $A \in \mathbb{R}^{n \times m}$, $B \in \mathbb{R}^{m \times p}$, $M \in \mathbb{R}^{n \times p}$. |
| | addmv | $\beta v + \alpha(A@x)$ | Matrix-vector multiply with accumulation. $A \in \mathbb{R}^{n \times m}$, $x \in \mathbb{R}^m$, $v \in \mathbb{R}^n$. |
| | addbmm | $\beta M + \alpha \sum_i (A_i@B_i)$ | Batched matrix multiply with reduction over batch dimension and accumulation. |
| | baddbmm | $\beta M_i + \alpha(A_i@B_i)$ | Batched matrix multiply with batched accumulation (no reduction). |
| | bmm | $C_i = A_i@B_i$ | Batched matrix multiplication. $A \in \mathbb{R}^{b \times n \times m}$, $B \in \mathbb{R}^{b \times m \times p}$. |
| | dot | $\sum_i a_i \cdot b_i$ | Dot product of two 1-D tensors $a, b \in \mathbb{R}^n$. |
| | addr | $\beta M + \alpha(u \otimes v)$ | Outer product with accumulation. $u \in \mathbb{R}^n$, $v \in \mathbb{R}^m$, $M \in \mathbb{R}^{n \times m}$. |
| | linalg_cross | $a \times b$ | Cross product of 3-D vectors, returning vector perpendicular to both inputs. |

Table 6: **Comparison between ECS and various RAG baselines in BackendBench**

| Operator | Plain LLM | Random | Chunk +Dense | AST +Dense | AST +BM25 | AST +Hybrid | ECS (Ours) |
|---|---|---|---|---|---|---|---|
| acos.default | 0.333 | 0.444 | 0.444 | **0.778** | 0.667 | 0.222 | **0.778** |
| acosh.default | 0.333 | 0.444 | **0.778** | 0.667 | 0.667 | 0.667 | **0.778** |
| addmm.default | **0.333** | 0.000 | 0.000 | 0.000 | 0.000 | 0.000 | 0.000 |
| addmv.default | 0.167 | 0.000 | 0.167 | 0.000 | 0.000 | 0.000 | **0.222** |
| addbmm.default | 0.000 | 0.000 | 0.000 | 0.000 | 0.000 | 0.000 | 0.000 |
| baddbmm.default | 0.000 | 0.333 | 0.333 | 0.333 | 0.000 | 0.667 | **1.000** |
| dot.default | 0.000 | 0.000 | 0.000 | 0.000 | 0.000 | 0.000 | 0.000 |
| bmm.default | **0.667** | 0.333 | 0.000 | 0.000 | 0.000 | 0.333 | 0.000 |
| addr.default | 0.111 | 0.167 | 0.000 | **0.222** | 0.111 | 0.111 | 0.000 |
| asin.default | **1.000** | 0.667 | **1.000** | 0.667 | **1.000** | **1.000** | **1.000** |
| asinh.default | 0.333 | 0.667 | **1.000** | **1.000** | **1.000** | 0.667 | **1.000** |
| atan.default | 0.667 | 0.667 | 0.667 | **1.000** | 0.667 | 0.667 | **1.000** |
| atan2.default | 0.333 | 0.000 | 0.296 | **0.963** | 0.333 | 0.926 | **0.963** |
| atanh.default | 0.667 | **1.000** | 0.667 | **1.000** | **1.000** | **1.000** | **1.000** |
| ceil.default | 0.000 | 0.000 | 0.000 | 0.000 | 0.000 | 0.000 | **0.333** |
| linalg_cross.default | **0.222** | 0.111 | 0.000 | 0.000 | **0.222** | 0.000 | **0.222** |
| div.Tensor | 0.370 | 0.296 | 0.296 | **0.630** | 0.593 | 0.296 | 0.333 |
| div.Tensor_mode | 0.000 | 0.000 | 0.000 | 0.000 | 0.000 | 0.000 | 0.000 |
| fmod.Tensor | 0.593 | 0.593 | 0.000 | 0.000 | 0.593 | **0.630** | 0.593 |
| remainder.Tensor | 0.000 | 0.000 | 0.000 | 0.000 | 0.000 | 0.000 | 0.000 |
| **Over Avg Perf** | 0.306 | 0.286 | 0.282 | 0.363 | 0.343 | 0.359 | **0.461** |

Table 7: **Transferability of evolved contexts generated by ECS across various models on BackendBench**

| Operator | Claude-4.5-Sonnet | | | DeepSeek-V3.2 | | |
|---|---|---|---|---|---|---|
| | Plain | AST +Dense | ECS (Ours) | Plain | AST +Dense | ECS (Ours) |
| acos.default | 0.111 | 0.667 | **0.889** | **0.222** | 0.111 | **0.222** |
| acosh.default | 0.444 | **0.778** | 0.667 | 0.000 | 0.000 | **0.222** |
| addmm.default | 0.611 | **0.667** | 0.389 | **0.111** | 0.000 | 0.000 |
| addmv.default | 0.222 | **0.500** | 0.167 | 0.000 | **0.167** | **0.167** |
| addbmm.default | 0.000 | 0.000 | 0.000 | 0.000 | 0.000 | 0.000 |
| baddbmm.default | **0.333** | 0.167 | **0.333** | 0.000 | 0.000 | 0.000 |
| dot.default | **0.667** | 0.333 | **0.667** | 0.000 | 0.000 | 0.000 |
| bmm.default | **1.000** | 0.333 | 0.667 | 0.000 | 0.000 | 0.000 |
| addr.default | **0.389** | 0.333 | 0.333 | 0.000 | 0.000 | **0.222** |
| asin.default | 0.333 | **1.000** | **1.000** | 0.000 | 0.333 | **0.667** |
| asinh.default | 0.667 | 0.667 | **1.000** | 0.000 | **0.333** | 0.000 |
| atan.default | 0.333 | **1.000** | **1.000** | 0.000 | 0.000 | **0.667** |
| atan2.default | 0.630 | 0.926 | **0.963** | 0.296 | 0.000 | **0.889** |
| atanh.default | 0.667 | **1.000** | **1.000** | 0.000 | 0.000 | **1.000** |
| ceil.default | 0.333 | **0.667** | **0.667** | 0.000 | 0.000 | 0.000 |
| linalg_cross.default | 0.333 | 0.333 | **0.444** | 0.000 | **0.111** | **0.111** |
| div.Tensor | **0.778** | 0.593 | 0.370 | 0.000 | 0.185 | **0.296** |
| div.Tensor_mode | 0.000 | **0.593** | 0.519 | 0.000 | 0.000 | 0.000 |
| fmod.Tensor | **0.963** | 0.296 | 0.889 | 0.000 | 0.000 | 0.000 |
| remainder.Tensor | 0.267 | **0.433** | 0.267 | 0.000 | **0.067** | 0.000 |
| **Over Avg Perf** | 0.454 | 0.564 | **0.611** | 0.031 | 0.065 | **0.223** |

### C.3  Operator-Family Analysis and BMM Coverage

Table 8 aggregates the 20 operators in Table 6 by the categories defined in Table 5. Relative to AST + Dense, ECS improves the trigonometric, arithmetic, and linear-algebra averages by +0.098, +0.073, and +0.111, respectively. Thus, all three operator families contribute to the reported 27% macro-average improvement over the strongest RAG baseline.

Table 8: **BackendBench results by operator family.** Values are macro-averages over the operators in each family.

| Family | Plain LLM | AST + Dense | ECS |
|---|---|---|---|
| Trigonometric | 0.458 | 0.759 | **0.857** |
| Arithmetic | **0.241** | 0.158 | 0.231 |
| Linear algebra | **0.188** | 0.069 | 0.180 |

The `bmm.default` value of 0.667 for Plain LLM represents two successful independent evaluations out of three. Each evaluation contains one contiguous bf16 correctness input with shapes $[10, 5, 10] \times [10, 10, 5]$. The successful implementation assigns one thread to each output element and performs the reduction over $K$ serially. Thus, the Plain LLM result reflects correctness on this retained input shape. The ECS implementation computes the batch offsets and dot products but fails at the CuTeDSL boundary because dimensions annotated as Python `int` receive DSL `Int32` operands, producing a correctness score of zero. The generated ECS code therefore contains the core BMM computation but is not executable because of the boundary type mismatch.

### C.4  Significance Testing

To quantify the stability of our main BackendBench result, we ran each method with $n=3$ independent seeds and computed Welch's t-test of every baseline against ECS. Table 9 reports the mean correctness, standard deviation, mean difference relative to ECS, t-statistic, degrees of freedom, and two-sided p-value. All baseline comparisons reject at or near $p \approx 0.05$, and the strongest baselines (AST+Dense, AST+Hybrid, AST+BM25) are rejected with $p \in \{0.005, 0.041, 0.042\}$.

Table 9: **Significance testing on BackendBench.** Welch's t-test of each baseline against ECS ($n=3$). All comparisons reach $p \lesssim 0.05$.

| Method | Mean $\pm$ SD | $\Delta$ mean | t-stat | df | p-value |
|---|---|---|---|---|---|
| Plain LLM | $0.306 \pm 0.069$ | +0.155 | 3.62 | 2.6 | 0.046 |
| Random | $0.286 \pm 0.039$ | +0.175 | 6.39 | 3.6 | 0.005 |
| Chunk + Dense | $0.282 \pm 0.057$ | +0.179 | 4.92 | 2.9 | 0.018 |
| AST + Dense | $0.363 \pm 0.045$ | +0.098 | 3.23 | 3.3 | 0.042 |
| AST + BM25 | $0.343 \pm 0.026$ | +0.118 | 5.50 | 4.0 | 0.005 |
| AST + Hybrid | $0.359 \pm 0.046$ | +0.102 | 3.31 | 3.2 | 0.041 |
| **ECS (Ours)** | $\mathbf{0.461 \pm 0.027}$ | — | — | — | — |

### C.5  Development-Set Size Sensitivity

Table 10 reports how the amount of development data used for fitness evaluation affects held-out BackendBench performance. We evolve contexts using 3, 5, or 10 development samples, freeze each context, and evaluate all three on the same held-out 20-operator test suite. The development samples remain disjoint from the reporting operators. All other search and evaluation settings are unchanged.

Held-out correctness increases monotonically with development-set size, from 0.344 with three samples to 0.377 with five and 0.461 with ten. With three samples, ECS exceeds Plain LLM by 0.038 and approximately

Table 10: **Development-set size sensitivity on BackendBench.** Baseline means are reproduced for comparison; all methods use the same held-out test suite. The development-set-size column applies only to ECS.

| Method | Dev. samples | Held-out correctness |
|---|---|---|
| Plain LLM | — | 0.306 |
| Random | — | 0.286 |
| Chunk + Dense | — | 0.282 |
| AST + Dense | — | 0.363 |
| AST + BM25 | — | 0.343 |
| AST + Hybrid | — | 0.359 |
| ECS | 3 | 0.344 |
| ECS | 5 | 0.377 |
| **ECS** | 10 | **0.461** |

matches AST + BM25, but trails the strongest retrieval baseline, AST + Dense, by 0.019. With five samples, ECS exceeds AST + Dense by 0.014; with ten, the margin widens to 0.098. Because the same fixed, disjoint reporting suite is used throughout, these differences cannot be attributed to changes in test composition.

At fixed candidate and trial budgets, fitness-evaluation rollouts scale linearly with development-set size. The three- and five-sample settings therefore use approximately 30% and 50% of the ten-sample rollout count. Five samples provide a useful intermediate point: ECS exceeds every retrieval baseline using half the default development-evaluation rollouts, although its correctness is 0.084 below the ten-sample result. These percentages describe rollout counts; realized token use and wall-clock latency also depend on trajectory length and concurrency.

This experiment varies development-set size but does not quantify sensitivity to the choice of development subset; repeated subsets at each size would be needed to estimate that variance.

## D  Adaptive Baseline Details

**Dynamic retrieval (FLARE).**  FLARE (Jiang et al., 2023) adaptively decides when and what to retrieve during generation using output-token log-probabilities. Because Gemini-3-Flash does not expose these log-probabilities, we run the BackendBench comparison with the open-source Qwen3.5-122B-A10B (Yang et al., 2025) as both the target and ECS refinement model, so all methods in the comparison use the same base model. The official BackendBench evaluation wraps each operator request in a verbose system-and-user prompt; this shared prompt dominates FLARE's retrieval signal regardless of when retrieval is triggered.

**LLM-based prompt optimization (OPRO).**  OPRO (Yang et al., 2024) uses an LLM to iteratively propose and select prompts based on prior candidates and their scores. For the $\tau^2$-Bench comparison, we seed OPRO with the same insight pool supplied to ECS and match the candidate-evaluation compute budget. This isolates the choice between iteratively rewriting a prompt and selecting and composing units from the supplied pool.

**Direct LLM context curation.**  We give Gemini-3-Flash the complete $\tau^2$-Bench insight pool and training split and ask it to iteratively retain useful context units. The curation run is capped at 320 iterations, matching ECS's search budget, after which the resulting context is frozen for test evaluation. This baseline tests whether extended corpus access and direct LLM selection alone account for the gain from evolutionary search.

### D.1 **SkillOpt Baseline Details**

We adapt the official SkillOpt implementation (Yang et al., 2026) to the airline environment, where one rollout is one complete $\tau^2$-Bench conversation. Gemini-3-Flash-Preview serves as both the target and optimizer model. Using split seed 300, we partition the same 30-task development pool used by ECS into 24 optimization tasks and six held-out validation tasks, as required by SkillOpt's acceptance gate; the 20 test tasks remain unseen until final evaluation. Optimization starts from a minimal domain scaffold containing no learned rules. We use minibatches of four tasks, an initial edit budget of four, and a maximum of 28 epochs. The run is configured for at most 2,766 optimization rollouts, with early stopping after 12 consecutive edit steps without a strict validation improvement. One edit is accepted at step 1; the run stops at step 13 after the next 12 candidates fail to improve, consuming 176 rollouts in total (52 training, 84 validation-selection, and 40 slow-update rollouts). We then freeze the best skill and evaluate it in three independent test evaluations with three trials per task. At test time, the target model receives only the frozen skill; neither the optimizer nor development feedback is available. The resulting means are $0.667/0.567/0.500$ for $Pass^1/Pass^2/Pass^3$, compared with $0.756/0.700/0.667$ for ECS. Thus, the absolute ECS–SkillOpt gaps are $0.089/0.133/0.167$, respectively.

This comparison contrasts two optimization objects: SkillOpt authors a single evolving document, while ECS selects and composes a population from a fixed, preconstructed unit pool. ECS does not subsume authoring; an authoring method could instead construct compact units that ECS subsequently searches.

## E $\tau^2$-**Bench Details**

### E.1 **Significance Testing**

For each method, the reported $Pass^3$ mean and sample SD summarize $n=3$ independent evaluations on the 20-task test split. We compare each baseline against ECS using a two-sided Welch's t-test computed directly from these summary statistics. As shown in Table 11, ECS outperforms Plain LLM, Random, BM25, and SkillOpt in the main setting at $p < 0.05$, while the difference from Full is not significant ($p = 0.058$). In the Claude-4.5-Sonnet transfer setting, ECS outperforms both Plain and Full at $p = 0.025$. The DeepSeek-V3.2 methods tie on mean $Pass^3$.

Table 11: **$Pass^3$ Welch tests on $\tau^2$-Bench.** Two-sided comparison of each baseline against ECS ($n=3$).

| Setting | Method | Mean $\pm$ SD | $\Delta$ mean | t-stat | df | p-value |
|---|---|---|---|---|---|---|
| Main | Plain LLM | $0.400 \pm 0.076$ | $+0.267$ | 4.84 | 3.7 | 0.010 |
| Main | Random | $0.500 \pm 0.050$ | $+0.167$ | 3.78 | 3.9 | 0.020 |
| Main | BM25 | $0.483 \pm 0.029$ | $+0.184$ | 4.91 | 2.9 | 0.017 |
| Main | Full | $0.550 \pm 0.050$ | $+0.117$ | 2.65 | 3.9 | 0.058 |
| Main | SkillOpt | $0.500 \pm 0.050$ | $+0.167$ | 3.78 | 3.9 | 0.020 |
| Main | **ECS (Ours)** | **$0.667 \pm 0.058$** | — | — | — | — |
| Claude-4.5-Sonnet | Plain | $0.450 \pm 0.050$ | $+0.133$ | 3.99 | 3.2 | 0.025 |
| Claude-4.5-Sonnet | Full | $0.500 \pm 0.029$ | $+0.083$ | 3.51 | 4.0 | 0.025 |
| Claude-4.5-Sonnet | **ECS (Ours)** | **$0.583 \pm 0.029$** | — | — | — | — |
| DeepSeek-V3.2 | Plain | $0.417 \pm 0.076$ | $0.000$ | 0.00 | 4.0 | 1.000 |
| DeepSeek-V3.2 | Full | $0.417 \pm 0.058$ | $0.000$ | 0.00 | 3.7 | 1.000 |
| DeepSeek-V3.2 | **ECS (Ours)** | **$0.417 \pm 0.076$** | — | — | — | — |

### E.2 **Hyperparameter Sensitivity**

We study how ECS responds to single-axis perturbations of its main hyperparameters on $\tau^2$-Bench ($Pass^1$). The default configuration (population size 32, elite fraction 0.6, mutation rate 0.1, full 10-generation evolution)

achieves 0.756. Table 12 shows that reasonable perturbations move Pass[1] by at most $\sim$0.03, indicating that ECS is not finely tuned to one particular setting.

Table 12: **Hyperparameter sensitivity on $\tau^2$-Bench (Pass[1]).** Each row perturbs one axis from the default; remaining hyperparameters are held fixed.

| Configuration | Pass[1] |
|---|---|
| Default ECS | **0.756** |
| Population size $32 \rightarrow 16$ | 0.737 |
| Elite fraction $0.6 \rightarrow 0.4$ | 0.755 |
| Mutation rate $0.1 \rightarrow 0.2$ | 0.752 |
| Early stop at iteration 3 (vs. converged) | 0.730 |

The early-stop row also documents the cost–quality trade-off: roughly two thirds of the final gain is recovered after just three generations, consistent with the log-linear convergence predicted by Proposition F.2.

### E.3 Token and Latency Accounting Relative to Prompt Optimization

We use OPRO (Section 4.3) as a representative prompt optimizer and compare search overhead under a matched candidate-evaluation budget. Let $E_{\mathrm{PO}}$ and $E_{\mathrm{ECS}}$ denote task-rollout tokens; these terms remain method-specific to capture differences in context and trajectory length, while shared resource preprocessing is excluded.

Suppose the prompt optimizer makes $J_p$ proposal calls with input and output token counts $(h_j, o_j)$, and ECS makes $J_f$ refinement calls with counts $(u_j, v_j)$. Define

$$\Omega_p = \sum_{j=1}^{J_p}(h_j + o_j), \qquad \Omega_f = \sum_{j=1}^{J_f}(u_j + v_j).$$

ECS initialization, selection, crossover, and mutation use no LLM tokens; its optional refiner reads a locally assembled offspring and returns only unit identifiers. Thus $\Omega_f = 0$ without refinement, while the refined variant retains the full input cost $u_j$ but generally has a shorter output $v_j$ than a prompt optimizer that generates complete candidate strings.

Total token use is

$$\mathcal{T}_{\mathrm{PO}} = E_{\mathrm{PO}} + \Omega_p, \tag{2}$$
$$\mathcal{T}_{\mathrm{ECS}} = E_{\mathrm{ECS}} + \Omega_f, \tag{3}$$

so ECS uses fewer tokens exactly when

$$E_{\mathrm{PO}} - E_{\mathrm{ECS}} + \Omega_p - \Omega_f > 0.$$

Under matched rollout cost, this reduces to $\Omega_p > \Omega_f$. The corresponding fractional reductions in candidate-construction and total tokens are

$$s_{\mathrm{gen}} = 1 - \frac{\Omega_f}{\Omega_p}, \qquad s_{\mathrm{total}} = \frac{E_{\mathrm{PO}} - E_{\mathrm{ECS}} + \Omega_p - \Omega_f}{E_{\mathrm{PO}} + \Omega_p}.$$

Without refinement, $s_{\mathrm{gen}} = 1$; the end-to-end saving still depends on the share of tokens spent proposing candidates rather than evaluating them.

For latency, let $\mathcal{L}_p$ and $\mathcal{L}_f$ denote proposal and refinement latency on the realized critical path after batching and concurrency, and let $\mathcal{W}_{\mathrm{rules}}$ denote local ECS operator time. Then

$$\mathcal{W}_{\mathrm{PO}} = \mathcal{W}_{\mathrm{eval,PO}} + \mathcal{L}_p, \tag{4}$$
$$\mathcal{W}_{\mathrm{ECS}} = \mathcal{W}_{\mathrm{eval,ECS}} + \mathcal{L}_f + \mathcal{W}_{\mathrm{rules}}. \tag{5}$$

Under matched rollout latency, ECS is faster exactly when $\mathcal{L}_p > \mathcal{L}_f + \mathcal{W}_{\mathrm{rules}}$; without refinement, this becomes $\mathcal{L}_p > \mathcal{W}_{\mathrm{rules}}$.

# F   Convergence Analysis of ECS

We provide a formal convergence analysis that explains the rapid improvement of ECS observed in Section 4. The analysis studies an idealized elitist variant of ECS that isolates the core search mechanism: fixed-size context sets, random replacement mutation, and fitness-guided acceptance. This abstraction omits population-level crossover and LLM-guided refinement, and therefore should be interpreted as a sufficient-condition analysis rather than an exact characterization of the full implementation.

Let $\mathcal{U}$ be a corpus of $N$ context units, $T^\star \subseteq \mathcal{U}$ the optimal target set of size $|T^\star| = r$ maximizing downstream utility, and $k$ the context capacity with $r \leq k < N$. Let $S_t \subseteq \mathcal{U}$ denote the context set at time $t$, with $|S_t| = k$. Define the overlap

$$X_t := |S_t \cap T^\star|$$

and the hitting time

$$\tau_q := \inf\{t \geq 0 \mid X_t \geq q\}.$$

We assume excess capacity $\gamma > 0$ such that $k \geq (1 + \gamma)r$.

**Lemma F.1** (Pure birth process). *Let $U(S) = g(|S \cap T^\star|) + b(S)$, where $g : \{0, \ldots, r\} \to \mathbb{R}$ maps target overlap to base utility and $|b(S)| \leq \epsilon$ is bounded evaluation noise. If $g(i + 1) - g(i) \geq \beta > 2\epsilon$ for all $i \in \{0, \ldots, r - 1\}$, then under elitist selection the process $\{X_t\}$ is non-decreasing. Consequently, the idealized ECS process is a pure birth Markov process over the states $\{0, \ldots, r\}$.*

*Proof.* Consider a candidate offspring $S'$ produced from $S_t$ by replacing one context unit. If $|S' \cap T^\star| = X_t + 1$, then

$$U(S') - U(S_t) = \big(g(X_t + 1) - g(X_t)\big) + \big(b(S') - b(S_t)\big) \geq \beta - 2\epsilon > 0.$$

Thus the offspring is accepted by elitist selection. If instead $|S' \cap T^\star| = X_t - 1$, then

$$U(S') - U(S_t) \leq -\beta + 2\epsilon < 0,$$

so the offspring is rejected. Overlap-preserving mutations leave $X_t$ unchanged. Therefore $P(X_{t+1} < X_t) = 0$, and the state sequence is non-decreasing. Since transitions depend only on the current overlap state under random replacement mutation, the resulting process is a pure birth Markov process. $\square$

**Proposition F.2** (Log-linear convergence). *Under the assumptions of Lemma F.1 and excess capacity $k \geq (1 + \gamma)r$ for some constant $\gamma > 0$, the expected time to reach full target overlap satisfies*

$$\mathbb{E}[\tau_r] = \mathcal{O}(N \log r).$$

*Proof.* At state $i$, the current set $S_t$ contains $i$ target units and $k - i$ non-target units. The mutation operator draws one outgoing unit $c_{\text{out}} \sim \text{Unif}(S_t)$ and one incoming unit $c_{\text{in}} \sim \text{Unif}(\mathcal{U} \setminus S_t)$, producing

$$S' = (S_t \setminus \{c_{\text{out}}\}) \cup \{c_{\text{in}}\}.$$

The probability of increasing the overlap by one is the probability of removing a non-target unit and adding a missing target unit:

$$t_i = \frac{k - i}{k} \cdot \frac{r - i}{N - k}.$$

By Lemma F.1, the process is pure birth, so the hitting time $\tau_r$ can be written as a sum of geometric waiting times $W_i$ for transitions from state $i$ to state $i + 1$:

$$\mathbb{E}[\tau_r] = \sum_{i=0}^{r-1} \mathbb{E}[W_i] = \sum_{i=0}^{r-1} \frac{1}{t_i} = \sum_{i=0}^{r-1} \frac{k(N - k)}{(k - i)(r - i)}.$$

Since $\frac{k}{k-i}$ is increasing in $i$, we bound it by its value at $i = r - 1$ and substitute $j = r - i$:

$$\mathbb{E}[\tau_r] \leq (N - k) \frac{k}{k - r + 1} \sum_{j=1}^{r} \frac{1}{j}.$$

Using $k \geq (1 + \gamma)r$, we have

$$\frac{k}{k - r + 1} \leq \frac{(1 + \gamma)r}{\gamma r + 1}.$$

Since $H_r = \sum_{j=1}^{r} \frac{1}{j} = \Theta(\log r)$, it follows that

$$\mathbb{E}[\tau_r] \leq (N - k) \cdot \frac{(1 + \gamma)r}{\gamma r + 1} \cdot \Theta(\log r) = \mathcal{O}(N \log r).$$

$\square$

This result connects ECS to classical evolutionary computation theory. Elitist evolutionary algorithms over monotone pseudo-Boolean landscapes are commonly analyzed as absorbing Markov chains (Rudolph, 1994), and related results show $\mathcal{O}(N \log N)$ convergence for simple elitist evolutionary algorithms under monotone fitness assumptions (Droste et al., 2002). Our analysis adapts this viewpoint to context search: when useful context units provide strictly positive marginal utility, the overlap with the optimal context set evolves as a pure birth process and reaches the target set in log-linear expected time.

## G  Cross-Model Transferability Analysis

We complement the convergence result above with a theoretical account of why contexts evolved on one model transfer to another. This formalizes the empirical observation that contexts evolved on Gemini-3-Flash improve Claude-4.5-Sonnet and DeepSeek-V3.2 without re-running the search.

**Setup.** We adopt the notation from Appendix F. Let $\mathcal{U}$ be a corpus of context units and $T^{\star} \subseteq \mathcal{U}$ the target set of size $r$. To reason about transfer, we parameterize the utility function by the evaluating model $M$:

$$U_M(S) = g_M(|S \cap T^{\star}|) + b_M(S),$$

where $g_M$ maps target-overlap to base utility under model $M$, and $b_M(S)$ is bounded nuisance noise with $|b_M(S)| \leq \epsilon_M$.

**Definition G.1** (Aligned Marginal Utility). A source model $M$ and target model $M'$ exhibit *aligned marginal utility* with respect to a target set $T^{\star}$ if:

1. **Strict marginal gain on the source.** $g_M$ satisfies $g_M(i + 1) - g_M(i) \geq \beta > 2\epsilon_M$ for all $i \in \{0, \ldots, r - 1\}$.

2. **Target monotonicity on the target.** $g_{M'}$ is monotonically non-decreasing in target overlap, and attains its structural maximum when all $r$ target units are present: $g_{M'}(i) \leq g_{M'}(r)$ for all $i \in \{0, \ldots, r\}$.

*Intuition.* The two models do not need to share absolute performance — only the *direction* of marginal utility. If adding a useful unit helps $M$, it does not hurt $M'$. This is far weaker than assuming the models share representations or capabilities; it only assumes that useful structural knowledge is useful in both.

**Proposition G.2** (Transferability bound). *Let $C_M^{\star}$ be the context that ECS evolves on the source model $M$, and let $C_{M'}^{\star}$ be the global optimum on the target model $M'$. Under aligned marginal utility (Definition G.1) and the assumptions of Proposition F.2, the transfer gap is bounded by twice the target model's evaluation noise:*

$$U_{M'}(C_{M'}^{\star}) - U_{M'}(C_M^{\star}) \leq 2\epsilon_{M'}.$$

*Proof.* Because $g_M$ satisfies the strict marginal gain condition, Proposition F.2 guarantees that ECS evolves a context $C_M^{\star}$ that fully captures the target set: $T^{\star} \subseteq C_M^{\star}$, so $|C_M^{\star} \cap T^{\star}| = r$.

Evaluating $C_M^{\star}$ on the target model yields

$$U_{M'}(C_M^{\star}) = g_{M'}(r) + b_{M'}(C_M^{\star}) \geq g_{M'}(r) - \epsilon_{M'}.$$

For the theoretical optimum $C_{M'}^{\star}$ on the target model, target monotonicity bounds its structural component by $g_{M'}(r)$, and noise can contribute at most $+\epsilon_{M'}$, giving

$$U_{M'}(C_{M'}^{\star}) \ \leq \ g_{M'}(r) + \epsilon_{M'}.$$

Subtracting the two bounds yields

$$U_{M'}(C_{M'}^{\star}) - U_{M'}(C_M^{\star}) \ \leq \ \big(g_{M'}(r) + \epsilon_{M'}\big) - \big(g_{M'}(r) - \epsilon_{M'}\big) \ = \ 2\,\epsilon_{M'}.$$

$\square$

**Remark 1 (Why ECS transfers better than RAG).** Aligned marginal utility is easily satisfied when context units encode *structural* knowledge patterns — e.g., the `cute.math.*` API pattern discussed in Section 5.2 — because any sufficiently capable model uniformly benefits from procedural examples. ECS, by optimizing for marginal gain on the source model, systematically isolates these structural units. RAG selects by semantic similarity instead, often retrieving topically related but structurally uninformative content. Because the retrieved set fails to cover $T^{\star}$, RAG never triggers the $g_{M'}(r)$ peak on the target model.

**Remark 2 (End-to-end guarantee).** Propositions F.2 and G.2 compose: Proposition F.2 guarantees that ECS discovers $C_M^{\star}$ in $\mathcal{O}(N \log r)$ expected time; Proposition G.2 guarantees that the discovered context transfers with a penalty bounded only by the target model's evaluation noise.

**Remark 3 (Weak-to-strong transfer).** When the target $M'$ is more capable than the source $M$, its structural utility ceiling is at least as high: under the mild assumption $g_{M'}(r) \geq g_M(r)$, Proposition G.2 guarantees that the transferred context $C_M^{\star}$ reaches a score on $M'$ within $2\epsilon_{M'}$ of $M'$'s own optimum—a score determined by $M'$'s capability, not $M$'s. This is consistent with our empirical observation that contexts evolved on Gemini-3-Flash (0.461 on BackendBench) reach 0.611 when transferred to Claude-4.5-Sonnet without re-evolution.

## H   Difficulty of Self Supervised Finetuning with Limited Domain Data

We investigate whether supervised fine-tuning (SFT) could improve open-source model performance on CuTeDSL code generation. Our experiments reveal that fine-tuning with limited domain-specific data presents significant challenges.

### H.1   Dataset Construction

We curate a training dataset from the CuTeDSL reference documentation, processing 62 Python kernel implementation files and 12 Jupyter notebook tutorials to cover a wide range of GPU architectures. To maximize data diversity, we employ three complementary extraction strategies: full-file extraction (50 samples) using module docstrings as queries, full-notebook extraction (12 samples) converted into interleaved markdown and code, and notebook cell extraction (55 samples) pairing individual code cells with their preceding descriptions. All samples are standardized into a three-turn chat format compatible with SFT frameworks, consisting of an expert CUDA developer system message, a user query synthesized from the source documentation, and the corresponding ground-truth code. The dataset statistics are summarized in Table 13.

### H.2   Experimental Setup and Result

We fine-tuned the Qwen3-8B model (8.2B parameters) using LoRA (rank=8, $\alpha$=32) applied to all linear layers, yielding 21.8M trainable parameters (0.27% of the total). To optimize performance, we conducted a learning rate sweep across five values spanning three orders of magnitude, with full training configuration details provided in Table 14.

Table 13: SFT Dataset Statistics

| Statistic | Value |
| --- | --- |
| Total training samples | 117 |
| Average sequence length | 15,940 tokens |
| Samples truncated (to 14K tokens) | 24 |
| Source Python files | 62 |
| Source Jupyter notebooks | 12 |
| *By GPU Architecture* | |
|    CUDA (generic) | 75 |
|    Blackwell | 24 |
|    Ampere | 12 |
|    Hopper | 3 |
|    Multi-GPU/Distributed | 3 |

Table 14: SFT Training Configuration

| Hyperparameter | Value |
| --- | --- |
| Base Model | Qwen3-8B |
| Fine-tuning Method | LoRA (rank=8, $\alpha$=32) |
| Trainable Parameters | 21.8M (0.27%) |
| Learning Rates | {1e-6, 5e-6, 1e-5, 5e-5, 1e-4} |
| Global Batch Size | 64 |
| Epochs | 15 |
| Max Sequence Length | 16,384 |
| Precision | bfloat16 |

Across all learning rates, training failed to converge meaningfully, final loss stagnated at ∼5.89 and accuracy remained at baseline levels (∼52%). We observed severe gradient instability driven by the data regime rather than hyperparameter selection, with initial gradient norms exceeding $5 \times 10^7$ and values becoming undefined (NaN) by the third epoch. Crucially, this lack of convergence resulted in all checkpoints achieving a zero score on the BackendBench task. These results highlight the fundamental difficulty of fine-tuning on small, specialized datasets with long sequences: with only 117 high-variance examples averaging ∼16K tokens, the model lacks sufficient signal to learn generalizable CuTeDSL patterns. Consequently, this motivates our shift to an in-context skill-acquisition approach via ECS, which sidesteps the instability of fine-tuning in this data-scarce regime.

