# OpenReview forum: "Evolutionary Context Search for Skill Acquisition"
_TMLR — Under review for TMLR_

### Review · Reviewer_KmG8 · 2026-06-23

**Summary Of Contributions:**

## Summary
The paper proposes ECS, a training-free method that reframes context construction as black-box optimization: text resources are split into "context units" (verbatim source, distilled insights, or Agent Skills), and a GA-style loop (dev-set fitness → elite selection → crossover → mutation → LLM refinement) searches for a context maximizing task accuracy. Headline results are a 27% relative gain over the best RAG baseline on BackendBench (CuTeDSL), a ~5% relative gain over Full-Context on $\tau^2$-bench airline, and cross-model transfer of contexts evolved on Gemini-3-Flash to Claude-4.5-Sonnet and DeepSeek-V3.2. Supporting material includes an idealized convergence analysis, a transferability bound, an ablation, and an appendix on SFT failure.

## Strength
- The core idea of optimizing corpus selection directly against task fitness rather than semantic similarity is clean, well-motivated, and of clear interest to TMLR's audience.
- Good experimental setting. Train/dev/test disjointness is documented; the refiner is the same model as the base, foreclosing the "smarter refiner did the work" confound; and Section 4.3 adds the right adaptive baselines (FLARE, OPRO, direct LLM curation) to make the evaluation more comprehensive.
- Appropriately-scoped theory. The convergence result, read as an idealized sufficient-condition analysis connecting ECS to classical (1+1)-EA theory, is a reasonable addition to the contribution.
## Limitation

My concerns are with the gap between several claims and the empirical evidences.
- **The improvement on BackendBench might be more narrow than the claim of "27\%".** Table 6 shows the improvement was carried by the trig/elementwise family, while the matmul family regresses below the Plain LLM: bmm 0.667→0.000, addmm 0.333→0.000, addr 0.111→0.000. However, the evolved context (Fig. 4) is 81% GEMM/MHA kernel code, meaning ECS's context damages the ops it matches. This also undercuts the Section 5.1 "coherent architectural stack" because the gains appear to come from one embedded pattern (cute.math.*), with the bulk of the context seems to be a distraction. The paper should report this rather and investigate in more details than presenting 0.461 as a uniform gain.
- **Transfer is overstated.** Table 2 shows ECS on DeepSeek-V3.2 underperforms Full on Pass@1 (0.600 vs 0.633) and underperforms Plain on Pass@2 — so "model-agnostic" holds only for Claude. The main paper's framing should match the Appendix's own aligned-marginal-utility caveat. Specifically, the main paper gives an impression that "ECS produces model-agnostic contexts", while the empirical results suggest that ECS transfers well to a capable model (Claude) on both tasks; transfers weakly or negatively to DeepSeek on the agentic task, consistent with the aligned-marginal-utility condition failing. However, this is only briefly mentioned in 1 sentence in the end of the main paper, while the detailed explanation is pushed to Appendix E.
- **The theory analyzes a different algorithm than the one implemented.** Prop. D.2 assumes strict elitism and abstracts away crossover/refinement, yielding a pure-birth process; while the implemented method uses fitness-proportional sampling + crossover, under which overlap can decrease. The monotone strict-marginal-gain assumption also conflicts with the paper's own "context distraction" findings. "The fitness curve is consistent with log-linear convergence" is unfalsifiable over five noisy generations. Prop. E.2 similarly predicts success only in cases that already worked (it assumes aligned marginal utility, which fails exactly where transfer fails). Both should be distinguished clearly as analogy unless validated on a synthetic task instantiating their assumptions.
- **Claim "SFT is fundamentally ineffective" is not well-supported.** Appendix F's SFT result NaN by epoch 3 with gradient norms >$5\times 10^7$, this seems like a setup/stability signature, not evidence that "SFT is fundamentally ineffective".

**Audience:**

Yes

**Audience Explanation:**

Yes, the topic of this work is highly relevant.

**Broader Impact Concerns:**

N /A

**Claims And Evidence:**

No

**Claims Explanation:**

The main paper might overclaim what the empirical results suggest (see my comments in the Weakness section).

**Requested Changes:**

I strongly suggest the authors to revise the claim to accurately reflect the empirical findings, specifically on these items:
- Analyze the gain and regression on BackendBench rather than stating 27\% as a uniform gain.
- ECS transfer positively is not a universal property across the tested LLMs. Please revise the claims accordingly.
- The theoretical analysis, although solid, does not match exactly with the implemented algorithm. Please state the differences and which assumption in the theory do not match with the implementation.
- The claim "SFT is fundamentally ineffective" needs to be revised or softened.

---

> ### Author Response · Authors · 2026-07-21
> **Response to Reviewer KmG8 - Part 1 of 3**
>
> We sincerely appreciate Reviewer KmG8’s thorough examination and recognition of our work as well-motivated. We have carefully addressed each point raised and implemented substantial revisions to enhance our manuscript accordingly.
>
> > The improvement on BackendBench might be more narrow than the claim of "27%". Table 6 shows the improvement was carried by the trig/elementwise family, while the matmul family regresses below the Plain LLM: bmm 0.667→0.000, addmm 0.333→0.000, addr 0.111→0.000. However, the evolved context (Fig. 4) is 81% GEMM/MHA kernel code, meaning ECS's context damages the ops it matches. This also undercuts the Section 5.1 "coherent architectural stack" because the gains appear to come from one embedded pattern (cute.math.*), with the bulk of the context seems to be a distraction. The paper should report this rather and investigate in more details than presenting 0.461 as a uniform gain.
> > Analyze the gain and regression on BackendBench rather than stating 27% as a uniform gain.
>
> We thank the reviewer for pointing out that the aggregate BackendBench score obscures important operator-family and operator-level heterogeneity. Section 4.2 (Main Results) now reports results by operator family instead of a plain unified score, while  Appendix C.3 (Operator-Family Analysis and BMM Coverage) provides the complete breakdown and a detailed analysis of bmm.
>
> Aggregating Table 6 by the three predefined operator families gives:
>
> | Family | Plain LLM | AST + Dense | ECS |
> |---|---:|---:|---:|
> | Trigonometric | 0.458 | 0.759 | 0.857 |
> | Arithmetic | 0.241 | 0.158 | 0.231 |
> | Linear algebra | 0.188 | 0.069 | 0.180 |
>
> The reported 27% improvement refers specifically to the relative improvement in the overall mean over AST + Dense Comparing to AST + Dense, ECS improves the trigonometric, arithmetic, and linear-algebra averages by 0.098, 0.073, and 0.111, respectively. Thus, the 27% aggregate improvement over the strongest RAG baseline is supported across all three families.
>
> Relative to Plain LLM, ECS improves the trigonometric-family average substantially, from 0.458 to 0.857, while the arithmetic and linear-algebra averages remain similar or slightly lower (0.241 to 0.231 and 0.188 to 0.180, respectively). At the operator level, ECS decreases from 0.667 to 0.000 on bmm, from 0.333 to 0.000 on addmm, and from 0.111 to 0.000 on addr. Conversely, it improves from 0.000 to 1.000 on baddbmm and from 0.167 to 0.222 on addmv, while remaining at 0.222 on linalg_cross. We now report these gains and regressions explicitly.
>
> We further investigated the bmm regression. The Plain LLM score of 0.667 represents two successful evaluations out of three on the only retained contiguous bf16 input configuration, with shapes [10,5,10] × [10,10,5]. The successful Plain LLM outputs assign one thread to each output element and perform the reduction over \(K\) serially. The ECS implementation generates the batch offsets and dot-product computation but fails at the CuTeDSL boundary because dimensions annotated as Python int receive DSL Int32 operands. Thus, the observed regression is genuine, although the failure is localized to interface-type handling. Because the benchmark retains only one input configuration for this operator, we do not generalize this diagnosis beyond the evaluated case. We retain the scores and report this analysis in Appendix C.3.
>
> Finally, we removed the “coherent architectural stack” interpretation. Figure 4 now describes the displayed snippets by their share of the selected context, Section 5.1 describes the selected kernel, GEMM/MHA, and interface material without inferring architectural coherence, and Section 5.2 presents cute.math.* only as a concrete cross-model transfer example. These revisions preserve the aggregate finding that ECS achieves 0.461 compared with 0.363 for the strongest RAG baseline, while making clear that this improvement is heterogeneous and accompanied by meaningful operator-level regressions.

---

> ### Author Response · Authors · 2026-07-21
> **Response to Reviewer KmG8 - Part 2 of 3**
>
> > Transfer is overstated. Table 2 shows ECS on DeepSeek-V3.2 underperforms Full on Pass@1 (0.600 vs 0.633) and underperforms Plain on Pass@2 — so "model-agnostic" holds only for Claude. The main paper's framing should match the Appendix's own aligned-marginal-utility caveat. Specifically, the main paper gives an impression that "ECS produces model-agnostic contexts", while the empirical results suggest that ECS transfers well to a capable model (Claude) on both tasks; transfers weakly or negatively to DeepSeek on the agentic task, consistent with the aligned-marginal-utility condition failing. However, this is only briefly mentioned in 1 sentence in the end of the main paper, while the detailed explanation is pushed to Appendix E.
> > ECS transfer positively is not a universal property across the tested LLMs. Please revise the claims accordingly.
>
> We have updated our manuscript to better clarify and sharpen the claims made regarding transfer, as suggested by the reviewer. We provide below a summary of these changes for convenience.
>
> We agree Table 2 shows positive transfer for Claude and only neutral/negative transfer for DeepSeek (though we note ECS delivers for DeepSeek almost the same performance as Full with a 6$\times$ token reduction) in Tau-bench, however we find in BackendBench (Figure 3) substantial positive transfer gain for DeepSeek. In BackendBench, ECS-discovered contexts transfer to DeepSeek to unlock a significant 7$\times$ performance gain over the plain baseline. Hence the sharpened claim is, as the reviewer notes, less broad than purely model-agnostic, but indeed broader than ‘only Claude’, as we do observe transfer yielding substantive performance improvements for DeepSeek in BackendBench.
>
> We note further that these differences in the transfer results across BackendBench and Tau-bench agree with the aligned marginal utility framework. Success in BackendBench requires precise and accurate knowledge of code structure and protocols, which are highly likely to be beneficial regardless of model capability or size – hence BackendBench is a prime candidate for aligned marginal utility. Tau-bench, by contrast, relies on discursive and conversational styles and adherence to less concrete policy guidelines, for which it is less obvious that the contexts evolved via a smaller, less capable model may necessarily transfer to larger, more capable models.
>
> We’ve therefore adjusted the transfer claims accordingly, highlighting ECS transfer is most likely to be model-agnostic in structured settings, such as BackendBench. Tau-bench is now presented as a probing, limiting case where aligned marginal utility is potentially less likely to hold, but where we find, encouragingly, that indeed strong transfer can still nonetheless take place.
>
>
> > The theoretical analysis, although solid, does not match exactly with the implemented algorithm. Please state the differences and which assumption in the theory do not match with the implementation.
>
> We thank the reviewer for the valuable feedback, we have adjusted our analysis correspondingly. Specifically, the changes are made for each following raised points.
>
>
> > Prop. D.2 assumes strict elitism and abstracts away crossover/refinement, yielding a pure-birth process; while the implemented method uses fitness-proportional sampling + crossover, under which overlap can decrease.
>
> We’ve adjusted the scope of our claim in Prop. D.2 as suggested, so that the proposition is framed around an abstraction of the core ECS accumulation mechanism, rather than the entire ECS mechanism complete with sampling, crossover, and refinement. We’ve therefore clarified that this result represents the convergence rate of the best-case core process, rather than the convergence rate for the entire ECS algorithm. We note that analyzing the convergence rate of the core mechanism under a stylized setting provides useful reference for the ECS mechanism’s convergence properties, as the core accumulation process is the main driver of the entire mechanism’s convergence. Additionally, the proposition shows that context search over the large $\binom{N}{K}$ combinatorial search space is reducible to log-linear time under fitness-guided search in the stylized monotone marginal utility setting. The additional empirical features of ECS, such as refinement, crossover and mutation, make the analysis less tractable, which is why we analyzed the isolated accumulation process. We therefore now state this modeling assumption more explicitly and reframe the result as a sufficient condition for the core ECS accumulation mechanism.

---

> > ### Author Response · Authors · 2026-07-21
> > **Response to Reviewer KmG8 - Part 3 of 3**
> >
> > > The monotone strict-marginal-gain assumption also conflicts with the paper's own "context distraction" findings.
> >
> > We thank the reviewer for pointing out this tension. We’ve duly clarified and narrowed the definition of monotone strict marginal gain to distinguish it from the context-distraction empirical findings. We’ve adjusted the setup in Proposition 3.1 to clarify that the strict marginal gain assumption applies to fixed context size, where replacing a non-target unit with a target unit improves utility under bounded distractor effects.
> >
> > >  "The fitness curve is consistent with log-linear convergence" is unfalsifiable over five noisy generations.
> >
> > We’ve softened this claim to note only that the fitness curve displays rapid empirical convergence, rather than the stricter, and indeed not yet proven, claim of log-linear convergence.
> >
> > >  Prop. E.2 similarly predicts success only in cases that already worked (it assumes aligned marginal utility, which fails exactly where transfer fails).
> >
> > We’ve clarified and tightened Proposition E.2 to be framed as a sufficiency condition under which transfer is expected to take place, as opposed to a more general predictor of when model pairs will transfer. We note framing this result as a sufficiency condition reveals the weak-to-strong generalization result that transfer does not depend on overall model capabilities being aligned in any sense. Instead, we require only the weak condition that models benefit from the same context units, where the extent or magnitude of that benefit also need not be aligned.
> >
> > > The claim "SFT is fundamentally ineffective" needs to be revised or softened.
> > > Claim "SFT is fundamentally ineffective" is not well-supported. Appendix F's SFT result NaN by epoch 3 with gradient norms > , this seems like a setup/stability signature, not evidence that "SFT is fundamentally ineffective".
> >
> > Thank you for pointing this out. We agree that the observed NaNs and large gradient norms  indicate instability in our specific training setup rather than a fundamental limitation of SFT. We have therefore revised “SFT is fundamentally ineffective” to “the evaluated SFT configurations were unstable and did not yield performance improvements in our data-scarce setting.”

---

### Review · Reviewer_tRCQ · 2026-06-29

**Summary Of Contributions:**

This paper proposes Evolutionary Context Search (ECS), a black-box method for constructing reusable context packages from external resources to improve downstream performance. Current methods typically use RAG to retrieve contexts relying on semantic similarity, which might lack precision. ECS converts documents, trajectories, or skills into context units, evaluates candidate context combinations on a small validation development set, and then applies evolutionary operators such as selection, crossover, mutation, and LLM-guided refinement to improve performance on the validation set. The method is evaluated on BackendBench/CuTeDSL code generation and $\tau^2$-Bench airline-domain agent tasks, with reported improvements over RAG, random context selection, full-context baselines, and several adaptive baselines.

**Audience:**

Yes

**Audience Explanation:**

Yes. The paper studies a timely and relevant problem for TMLR readers: how to adapt LLMs to new tasks or domains using external resources without weight updates. The idea that downstream task performance can be a better objective than semantic similarity for context selection is practically useful. ECS is also simple, black-box, and potentially applicable to closed-source models.

Even if the current submission is not yet convincing enough for acceptance, the findings would likely interest researchers working on RAG, prompt optimization, in-context learning, LLM agents, and post-deployment model adaptation. The qualitative analysis showing that evolved context can transfer structural API patterns, such as ``cute.math.*`` usage in CuTeDSL, is especially useful because it illustrates why similarity-based retrieval can miss task-critical information.

**Broader Impact Concerns:**

I have no concerns on the ethical implications of this work.

**Claims And Evidence:**

Yes

**Claims Explanation:**

The paper provides reasonably convincing evidence for its main empirical claims. Figure 2 shows that ECS improves BackendBench correctness from 0.363 for the strongest RAG baseline, AST+Dense, to 0.461, and Table 1 shows consistent gains on $\tau^2$-Bench over the Full Context baseline. The transfer claim is also supported: Figure 3 shows that contexts evolved with Gemini-3-Flash improve both Claude-4.5-Sonnet and DeepSeek-V3.2 on BackendBench, and Table 2 shows similarly positive or competitive transfer on τ²-Bench, especially for Claude-4.5-Sonnet and with substantially fewer tokens than Full Context. The paper further strengthens the evidence with adaptive baselines in Tables 3 and 4, and with ablations in Figure 6 showing that fitness-guided selection and mutation are important components of ECS.

However, I think the evidence could still be strengthened in a few ways. The evaluation is based on two main domains, so additional benchmarks like TerminalBench or Claw-Eval-Lite would better support the broader framing of ECS as a *general skill-acquisition* method. Besides, since ECS optimizes directly on a small development set, I would also like to see more analysis of dev-set sensitivity, such as multiple development splits or a dev-set-size scaling study. Finally, $\tau^2$-Bench would benefit from confidence intervals or significance tests similar to the BackendBench analysis in Appendix B.3.

**Requested Changes:**

As mentioned above,

- Add dev-set sensitivity analysis. The current disjoint train/dev/test setup is reassuring, but the evidence would be stronger with multiple development splits, a dev-set-size scaling study, or an experiment where contexts evolved on different dev subsets are evaluated on the same held-out test set.
- Significance tests for $\tau^2$-Bench.
- More evaluation scenarios.

What's more,

- Speed analysis / token usage analysis. From my perspective, initialization, selection, mutation and crossover operators are rule-based compared to prompt optimization methods like GEPA, which might reduce resource for better performance.
- One question: Can ECS be combined with current harness framework like Codex?
- Could you please compare ECS with recently skill evolution based methods like SkillOpt? For example, let the dev-set be the training set in these work. This may strengthen the claim in this paper.
- Typos in the main text:
    - p1, last para, "text-based prompt augmentations offers" -> "text-based prompt augmentations offer";
    - p2, contribution 1, "seleciton" -> "selection";
    - p4, "Insights." part, "we prompt another model to analyze the errors and extracts rules" -> "we prompt another model to analyze the errors and extract rules";
    - p5, "Initialization." part, "while task with short units" -> "while tasks with short units";
    - p8, LHS of Fig. 3, "capapbilities" -> "capabilities";
    - p8, the line above "Observation 3", "as an data curation process" -> "as a data curation process";
    - Caption in Figure 5, "Quantitative results show evolved context" -> "Quantitative results show that the evolved context".

  A proofread is needed.

Overall, I think the motivation behind this paper is good and the method is general. I would like to see the revised version.

---

> ### Author Response · Authors · 2026-07-21
> **Response to Reviewer tRCQ - Part 1 of 3**
>
> We deeply appreciate Reviewer tRCQ’s thorough examination and valuable advice. We have carefully addressed each point raised and implemented substantial revisions to enhance our manuscript accordingly.
>
> > The evaluation is based on two main domains, so additional benchmarks like TerminalBench or Claw-Eval-Lite would better support the broader framing of ECS as a general skill-acquisition method
> >  More evaluation scenarios.
>
> We thank the reviewer for this suggestion. We have added Terminal-Bench 2.1 as a third evaluation setting.
>
>
> We use the benchmark's 89 tasks to construct a fixed category- and difficulty-stratified split of 53 training and 36 held-out test tasks. For each training task, we collect one Gemini-3-Flash baseline trajectory and extract one independently selectable insight, yielding a 53-unit pool. ECS uses a fixed ten-task subset of the training split for fitness evaluation, with five generations, a population of eight, an elite fraction of 0.5, a ten-unit context cap, and no LLM refinement. The evolved context is frozen before test evaluation. BM25 RAG and Random draw only from the same training-derived pool and use the same maximum context-unit budget; no test task or trajectory is used for insight construction, search, or baseline setup. All methods use Gemini-3-Flash with the Terminus-2 agent and are evaluated three times on the same fixed test split.
>
> The new results are:
>
> | Method | Held-out accuracy |
> |---|---:|
> | Plain LLM | $0.519 \pm 0.016$ |
> | BM25 RAG | $0.537 \pm 0.016$ |
> | Random | $0.556 \pm 0.056$ |
> | **ECS** | **$0.611 \pm 0.028$** |
>
> ECS achieves the highest mean held-out accuracy, with absolute gains of approximately 0.056 over Random, 0.074 over BM25 RAG, and 0.093 over Plain LLM. This result extends the evidence for ECS beyond the original coding-DSL and conversational-policy domains to heterogeneous terminal tasks. Because the comparison uses three runs on one fixed split, we treat it as additional evidence across a broader range of scenarios rather than as proof of universal superiority.
>
> The revised manuscript reports the benchmark description and configuration in Section 4.1, the results and comparison table in Section 4.2 (Table 1), the complete split and contamination controls in Appendix A, and corresponding updates to the Abstract and Introduction.
>
>
> > I would also like to see more analysis of dev-set sensitivity, such as multiple development splits or a dev-set-size scaling study
> > Add dev-set sensitivity analysis. The current disjoint train/dev/test setup is reassuring, but the evidence would be stronger with multiple development splits, a dev-set-size scaling study, or an experiment where contexts evolved on different dev subsets are evaluated on the same held-out test set.
>
> We thank the reviewer for this helpful suggestion. We added a BackendBench development-set-size scaling study in Appendix C.5 and a pointer in Section 4.1. We evolve separate contexts using 3, 5, or 10 development samples, freeze each context, and evaluate all three on the same held-out 20-operator test suite. The development samples remain strictly disjoint from the reporting operators, and all other search and evaluation settings are unchanged.
>
> We report the size-scaling results together with the existing BackendBench baselines for context:
>
> | Method | ECS development samples | Held-out correctness |
> |---|---:|---:|
> | Plain LLM | -- | 0.306 |
> | Random | -- | 0.286 |
> | Chunk + Dense | -- | 0.282 |
> | AST + Dense | -- | 0.363 |
> | AST + BM25 | -- | 0.343 |
> | AST + Hybrid | -- | 0.359 |
> | ECS | 3 | 0.344 |
> | ECS | 5 | 0.377 |
> | **ECS** | **10** | **0.461** |
>
> Held-out correctness increases monotonically from 0.344 with three samples to 0.377 with five and 0.461 with ten. With three samples, ECS exceeds Plain LLM by 0.038 and approximately matches AST + BM25, but trails the strongest retrieval baseline, AST + Dense, by 0.019. With five samples, ECS exceeds AST + Dense by 0.014; with ten, the margin widens to 0.098.
>
> Because all three contexts are evaluated on the same fixed, disjoint reporting suite, this trend cannot be attributed to changes in test composition. The performance gain comes with a proportional increase in search compute. At fixed candidate and trial budgets, fitness-evaluation rollouts scale linearly with development-set size, so the three- and five-sample settings use approximately 30% and 50% of the ten-sample rollout count. Five samples therefore provide a useful intermediate point: ECS exceeds every retrieval baseline using half the default development-evaluation rollouts, although its correctness is 0.084 below the ten-sample result. These percentages describe rollout counts; realized token use and wall-clock latency also depend on trajectory length and concurrency.

---

> ### Author Response · Authors · 2026-07-21
> **Response to Reviewer tRCQ - Part 2 of 3**
>
> > Bench would benefit from confidence intervals or significance tests similar to the BackendBench analysis in Appendix B.3.
> > Significance tests for tau-Bench.
>
> We thank the reviewer for this suggestion.  The updated Table 11 (also listed below) now reports the mean and standard deviation over three runs, together with two-sided Welch’s t-tests on Pass^3. In the main setting, ECS performs significantly better than Plain LLM, Random, BM25, and SkillOpt. Compared with Full Context, ECS achieves an absolute gain of (0.117), the difference (p=0.058) narrowly misses the conventional significance threshold. The transfer results also vary across models: ECS yields significant gains with Claude-4.5-Sonnet, whereas it ties the baselines on Pass^3 with DeepSeek-V3.2. We have revised the manuscript to state these results and limitations explicitly.
>
> | Setting | Method | Pass$^3$ mean $\pm$ SD | Difference vs. ECS | $p$-value |
> |---|---|---:|---:|---:|
> | Main | Plain LLM | $0.400 \pm 0.076$ | 0.267 | 0.010 |
> | Main | Random | $0.500 \pm 0.050$ | 0.167 | 0.020 |
> | Main | BM25 | $0.483 \pm 0.029$ | 0.184 | 0.017 |
> | Main | Full Context | $0.550 \pm 0.050$ | 0.117 | 0.058 |
> | Main | SkillOpt | $0.500 \pm 0.050$ | 0.167 | 0.020 |
> | Main | **ECS (Ours)** | **$0.667 \pm 0.058$** | -- | -- |
> | Claude-4.5-Sonnet | Plain | $0.450 \pm 0.050$ | 0.133 | 0.025 |
> | Claude-4.5-Sonnet | Full Context | $0.500 \pm 0.029$ | 0.083 | 0.025 |
> | Claude-4.5-Sonnet | **ECS (Ours)** | **$0.583 \pm 0.029$** | -- | -- |
> | DeepSeek-V3.2 | Plain | $0.417 \pm 0.076$ | 0.000 | 1.000 |
> | DeepSeek-V3.2 | Full Context | $0.417 \pm 0.058$ | 0.000 | 1.000 |
> | DeepSeek-V3.2 | **ECS (Ours)** | **$0.417 \pm 0.076$** | -- | -- |
>
>
> > Speed analysis / token usage analysis. From my perspective, initialization, selection, mutation and crossover operators are rule-based compared to prompt optimization methods like GEPA, which might reduce resource for better performance.
>
> We thank the reviewer for identifying this potential efficiency advantage. There indeed is such an advantage and we have added an analysis correspondingly.
>
> In the revised paper, Section 5.4 now summarizes the resource comparison, while Appendix E.3 provides the matched-budget derivation, resource table, latency condition, and BackendBench case study..
>
> The analysis separates the task-rollout cost shared by both approaches from the method-specific cost of constructing candidates. For a matched candidate-evaluation budget, let $E_{\mathrm{PO}}$ and $E_{\mathrm{ECS}}$ denote rollout tokens, $\Omega_p$ the tokens used by prompt-optimizer proposal calls, and $\Omega_f$ those used by ECS refinement calls. Total token usage is
>
> $$
> \mathcal{T}\_{\mathrm{PO}} = E\_{\mathrm{PO}} + \Omega\_p, \qquad
> \mathcal{T}\_{\mathrm{ECS}} = E\_{\mathrm{ECS}} + \Omega\_f .
> $$
> ECS's rule-based initialization, selection, crossover, and mutation use no LLM calls or tokens. Its optional refiner reads a locally assembled offspring and returns only unit identifiers to keep; $\Omega_f$ includes both this call's input and output tokens. Therefore, under matched rollout costs, ECS uses fewer tokens exactly when $\Omega_p>\Omega_f$.
>
> For latency, after accounting for batching and concurrency, ECS is faster when proposal latency exceeds refinement latency plus local operator time.
>
> On BackendBench, removing refinement changes correctness only from $0.461$ to $0.458$; candidate construction then becomes fully local and eliminates all proposal-model calls and tokens for a $0.003$ absolute performance change. When refinement is beneficial, the end-to-end advantage depends on refinement inputs, proposal strategy, realized call counts, rollout costs, and concurrency.
>
> We use OPRO for the empirical comparison because it is evaluated in our experiments: under the matched evaluation budget, ECS achieves $0.756/0.700/0.667$ versus OPRO's $0.694/0.578/0.500$ on Pass$^1$/Pass$^2$/Pass$^3$ of $\tau^2$-Bench. The accounting applies to GEPA-like methods that use LLM calls to propose updates, but we do not claim a direct measurement of GEPA's token usage or runtime.
>
> > One question: Can ECS be combined with current harness framework like Codex?
>
> Yes. ECS is a general approach for searching and optimizing the context and is largely orthogonal to the underlying task-solving harness and could be integrated with an existing harness such as Codex. In this setup, Codex would serve as the execution scaffold for solving tasks and interacting with tools, while ECS would operate as an outer-loop context optimization method. It could use Codex’s performance on a development set as the fitness signal to search for and evolve useful context combinations derived from the available corpus.

---

> > ### Author Response · Authors · 2026-07-21
> > **Response to Reviewer tRCQ - Part 3 of 3**
> >
> > > Could you please compare ECS with recently skill evolution based methods like SkillOpt? For example, let the dev-set be the training set in these work. This may strengthen the claim in this paper.
> >
> > We thank the reviewer for this suggestion. We have added SkillOpt[1] as a baseline on the airline domain of $\tau^2$-Bench. The new comparison is reported in Section 4.2 and Table 4, with implementation and budget details in Appendix D.1.
> >
> > We adapted the github official implementation[2] and used Gemini-3-Flash-Preview as both the target and optimizer model. Following the reviewer's suggestion, SkillOpt draws its optimization data from the same 30-task development pool used by ECS: 24 tasks are used for optimization and six for held-out validation, as required by SkillOpt's acceptance gate. The 20 test tasks remain unseen during optimization, and the best skill is frozen before test evaluation. Appendix D.1 reports the configured maximum of 2,766 optimization rollouts.
> >
> > The updated results are:
> >
> > | Method | Pass$^1$ | Pass$^2$ | Pass$^3$ |
> > |---|---:|---:|---:|
> > | Plain LLM | 0.622 | 0.489 | 0.400 |
> > | SkillOpt | 0.667 | 0.567 | 0.500 |
> > | ECS | 0.756 | 0.700 | 0.667 |
> >
> > SkillOpt improves over the plain model at every Pass level, while ECS is higher than SkillOpt by $0.095/0.172/0.217$ on Pass$^1$/Pass$^2$/Pass$^3$. Across three independent test evaluations, Pass$^3$ is $0.500\pm0.050$ for SkillOpt and $0.667\pm0.058$ for ECS (two-sided Welch test, $p=0.008$).
> >
> > This comparison contrasts SkillOpt's optimization of a single authored skill document with ECS's selection and composition of units from a preconstructed pool. We therefore interpret the result as evidence for ECS relative to this evaluated skill-optimization baseline, not as universal superiority over skill-authoring methods; the two mechanisms may also be complementary.
> >
> > [1] Y. Yang et al., “SkillOpt: Executive strategy for self-evolving agent skills,” arXiv preprint arXiv:2605.23904, 2026.
> >
> > [2] https://github.com/microsoft/SkillOpt
> >
> > > Typos in the main text:
> > p1, last para, "text-based prompt augmentations offers" -> "text-based prompt augmentations offer";
> > p2, contribution 1, "seleciton" -> "selection";
> > p4, "Insights." part, "we prompt another model to analyze the errors and extracts rules" -> "we prompt another model to analyze the errors and extract rules";
> > p5, "Initialization." part, "while task with short units" -> "while tasks with short units";
> > p8, LHS of Fig. 3, "capapbilities" -> "capabilities";
> > p8, the line above "Observation 3", "as an data curation process" -> "as a data curation process";
> > Caption in Figure 5, "Quantitative results show evolved context" -> "Quantitative results show that the evolved context".
> >
> > Thank you for pointing out these issues, we have corrected the typos in the revision correspondingly.

---

### Review · Reviewer_Pvvj · 2026-07-02

**Summary Of Contributions:**

The paper proposes Evolutionary Context Search, a black box procedure for turning an external corpus into a static reusable context. The authors frame context construction as task driven optimization: candidate contexts are evaluated on a small development set, and an evolutionary loop selects, recombines, mutates, and refines them. The goal is to give a deployed language model new domain capability without updating its weights.

The core idea is useful and timely. The paper makes a clear case that semantic similarity is often a weak proxy for whether a piece of context will actually change model behavior. The qualitative examples are helpful, especially the code setting where an evolved context teaches a structural API pattern that transfers across models.

The main weakness is that the evidence supports a narrower claim than the framing suggests. The comparisons are mostly against static or lightly adaptive retrieval and curation methods. Since the output of ECS is itself a static reusable artifact, the most direct baseline is an automatic offline skill or memory creation method that reads the same corpus, uses the same development feedback, and writes a reusable artifact. That comparison is missing.

**Audience:**

Yes

**Audience Explanation:**

Yes. Many readers would care about methods that adapt deployed models through context alone. The idea is likely to interest researchers working on retrieval, prompting, agents, and model adaptation.

**Broader Impact Concerns:**

I do not see major broader impact concerns beyond the usual risks of improving automated code generation and policy following agents.

**Claims And Evidence:**

No

**Claims Explanation:**

The experiments support the claim that ECS is a promising offline context curation method. The authors show gains over several retrieval baselines, provide useful ablations, and give interesting evidence that static contexts can transfer across models.

The broader claim about skill acquisition is less well supported. The paper argues that ECS moves beyond retrieval and opens a path for automated context discovery, yet the strongest natural competitor for the same final product is absent. A simple baseline would let the same model explore the corpus with tools such as file listing, search, grep, and file reading, then write a static skill or memory. A stronger version would evaluate that artifact on the same development set, inspect failures, and revise it for a few rounds. At test time, only the final artifact would be supplied, with no tools.

This baseline is especially important for the code benchmark, where the knowledge source is already a repository. It directly tests whether evolutionary selection is needed, or whether an agent can read the files and distill compact rules on its own. For example, the central API pattern highlighted in the paper could plausibly be written as a short rule by an offline skill creation procedure, rather than recovered through selecting a long source file.

There are also a few narrower evidential gaps. Full Context is discussed as a baseline. It is not reported for BackendBench, where it would be informative. Some transfer claims should be stated more carefully, since the tau2 Bench result with DeepSeek is mixed. The statistical support is also thin in places, with only a small number of seeds for the main code result and no comparable uncertainty reporting for the tau2 Bench table.

Overall, the paper has an interesting result if framed as offline static context optimization. The current evidence does not yet establish the stronger claim that the evolutionary procedure is the right general mechanism for automated skill acquisition from a corpus.

**Requested Changes:**

1. Add an offline agentic skill or memory creation baseline. The baseline should receive the same corpus and development tasks, explore the corpus with simple tools, write a static artifact, and use only that artifact at test time. The budget should be matched to ECS in terms of model calls, read tokens, development evaluations, and final context length. A feedback driven version that revises the artifact after development failures is the most important comparison.

2. Report Full Context on BackendBench, or explain why it is infeasible. This baseline is defined in the experimental setup and used elsewhere, yet it is absent from the main code benchmark where it would clarify how much long context alone can recover.

3. Narrow the claims if the new baseline is not added. The current evidence supports ECS as an effective offline context curation method compared with standard retrieval baselines. It does not yet show that evolutionary search is superior to automatic skill induction or agentic corpus exploration.

4. Strengthen the uncertainty reporting. The main BackendBench result uses a small number of seeds, and the strongest comparisons are not overwhelmingly separated. The tau2 Bench results should also include uncertainty estimates. Transfer claims should report absolute numbers and mixed cases as plainly as relative improvements.

---

**Additional Suggestions**

Clarify the distinction between selecting existing skill units and creating a new skill from a corpus. This matters because an authoring baseline can compress a large source file into a short rule, while ECS is largely extractive over its unit pool.

The paper would also benefit from a clearer statement of its best applicability regime. ECS seems most compelling when the domain is stable, the development set is representative, an automatic metric is available, and the resulting artifact will be reused enough to amortize the search.

---

> ### Author Response · Authors · 2026-07-21
> **Response to Reviewer Pvvj - Part 1 of 3**
>
> We sincerely appreciate Reviewer Pvvj’s thorough evaluation and recognition of our work’s usefulness. Following the valuable feedback, we have enhanced our manuscript for a cleaner scope and richer empirical findings. We address each point raised below. Revisions to our manuscript are shown in blue.
>
> > Add an offline agentic skill or memory creation baseline. The baseline should receive the same corpus and development tasks, explore the corpus with simple tools, write a static artifact, and use only that artifact at test time. The budget should be matched to ECS in terms of model calls, read tokens, development evaluations, and final context length. A feedback driven version that revises the artifact after development failures is the most important comparison.
>
> Thank you for this important suggestion. Following the reviewer’s suggestion, we have added SkillOpt [1], a recent open-source method for feedback-driven optimization of reusable agent skills. SkillOpt iteratively converts scored success and failure trajectories from dev set into bounded edits to a single skill document, and uses the final artifact at test time.
>
> We adapted its official implementation [2] on Tau2-bench. This baseline receives the same source corpus and dev tasks as ECS, iteratively constructs and revises a static skill artifact with the dev set feedback. We match the methods in terms of the model call budgets, development evaluations, and maximum final artifact length; the exact budget accounting is reported in Appendix D.1.
>
> The empirical result in the updated table 4 (also listed below) suggests that SkillOpt improves over the plain model. ECS achieves higher performance in the matched setting. We interpret this as evidence for ECS’s effectiveness relative to this particular skill-optimization baseline, rather than as establishing universal superiority over all agentic skill-induction or corpus-exploration procedures.
>
> | Method | Pass$^1$ | Pass$^2$ | Pass$^3$ |
> |---|---:|---:|---:|
> | Plain LLM | 0.622 | 0.489 | 0.400 |
> | SkillOpt | 0.667 | 0.567 | 0.500 |
> | ECS | 0.756 | 0.700 | 0.667 |
>
> We have also revised the manuscript to avoid the broader claim that evolutionary search is the universally preferred mechanism for skill induction from arbitrary corpora. Our revised claim is that ECS is an effective method for offline, context-mediated skill acquisition and outperforms the retrieval, prompt-optimization, and skill-curation baselines evaluated in our experiments. And agentic skill-authoring methods may also be complementary: they can synthesize candidate skill units from a corpus, while ECS can subsequently select and compose those units according to downstream task performance.
>
> [1] Y. Yang et al., “SkillOpt: Executive strategy for self-evolving agent skills,” arXiv preprint arXiv:2605.23904, 2026.
>
> [2] https://github.com/microsoft/SkillOpt

---

> ### Author Response · Authors · 2026-07-21
> **Response to Reviewer Pvvj - Part 2 of 3**
>
> > Report Full Context on BackendBench, or explain why it is infeasible. This baseline is defined in the experimental setup and used elsewhere, yet it is absent from the main code benchmark where it would clarify how much long context alone can recover.
>
> We thank the reviewer for this suggestion. A literal Full Context baseline is infeasible on BackendBench because the complete context pool contains approximately 1.2M tokens, exceeding Gemini-3-Flash's 1,024k-token context window. We agree, however, that a near-full long-context comparison is informative. We therefore added a Near-Full-Context baseline that randomly removes only as many context units as needed to fit within the model's limit. The resulting context contains 1,003.5k tokens, leaving the remaining window for the task instructions and query.
>
> We have included this new result to our Figure 2. And the updated BackendBench results are:
>
> | Method | Correctness rate | Context tokens (k) |
> |---|---:|---:|
> | Near-Full-Context | 0.383 | 1,003.5 |
> | Plain LLM | 0.306 | 0 |
> | Random | 0.286 | 30.3 |
> | ECS | **0.461** | 49.2 |
>
> Near-Full-Context improves over the Plain LLM by 0.077 absolute correctness, showing that broad corpus coverage is useful. It is still inferior to ECS even though it consumes  20.4 times as many context tokens (1,003.5k versus 49.2k). We now report the construction, context length, and result of this baseline in the experimental setup and main BackendBench results, with all corresponding manuscript revisions shown in blue. We also explicitly call it Near-Full-Context rather than Full Context so that the comparison does not imply that all 1.2M corpus tokens were included.
>
>
> > Narrow the claims if the new baseline is not added. The current evidence supports ECS as an effective offline context curation method compared with standard retrieval baselines. It does not yet show that evolutionary search is superior to automatic skill induction or agentic corpus exploration.
>
> We have revised this section to increase clarity. In the revision, we both add a relevant feedback-driven skill-authoring comparison and narrow the claims. As described above, SkillOpt learns a reusable static skill from development feedback and is therefore a stronger comparison than retrieval alone.
>
> Accordingly, the revised manuscript makes only a bounded empirical claim: ECS is an effective offline static-context optimization method that outperforms the retrieval, prompt-optimization, direct-curation, and SkillOpt baselines evaluated in our experiments. We do not claim that evolutionary search is universally preferable to automatic skill induction or agentic corpus exploration. We also describe the mechanisms as potentially complementary: an authoring method can synthesize compact candidate skill units from a corpus, while ECS can select and compose those units using downstream task performance.
>
> We revised this scope consistently in blue throughout the paper. In particular, the Abstract now states that the results support ECS ``a practical approach to post-deployment skill acquisition'' and do not establish universal superiority over agentic skill authoring. Section 2 (Related Work) distinguishes search over a supplied resource pool from methods that author new skills. Section 4.3 (Beyond Static Retrieval) adds SkillOpt and limits the empirical conclusion to the adaptive baselines evaluated in our experiments. Finally, Section 6 (Conclusions) interprets the evidence as support for offline context optimization relative to the evaluated baselines rather than universal superiority over skill-authoring or corpus-exploration methods.

---

> > ### Author Response · Authors · 2026-07-21
> > **Response to Reviewer Pvvj - Part 3 of 3**
> >
> > > Strengthen the uncertainty reporting. The main BackendBench result uses a small number of seeds, and the strongest comparisons are not overwhelmingly separated. The tau2 Bench results should also include uncertainty estimates. Transfer claims should report absolute numbers and mixed cases as plainly as relative improvements.
> >
> > Thank you for highlighting the need for stronger uncertainty reporting. In response,  we added means and sample standard deviations over three independent evaluations and two-sided Welch tests for Pass^3 of Tau2-bench in Table 9. The new results are summarized below; the difference column reports ECS minus the comparison method.
> >
> > | Setting | Method | Pass$^3$ mean $\pm$ SD | Difference vs. ECS | $p$-value |
> > |---|---|---:|---:|---:|
> > | Main | Plain LLM | $0.400 \pm 0.076$ | 0.267 | 0.010 |
> > | Main | Random | $0.500 \pm 0.050$ | 0.167 | 0.020 |
> > | Main | BM25 | $0.483 \pm 0.029$ | 0.184 | 0.017 |
> > | Main | Full Context | $0.550 \pm 0.050$ | 0.117 | 0.058 |
> > | Main | SkillOpt | $0.500 \pm 0.050$ | 0.167 | 0.020 |
> > | Main | **ECS (Ours)** | **$0.667 \pm 0.058$** | -- | -- |
> > | Claude-4.5-Sonnet | Plain | $0.450 \pm 0.050$ | 0.133 | 0.025 |
> > | Claude-4.5-Sonnet | Full Context | $0.500 \pm 0.029$ | 0.083 | 0.025 |
> > | Claude-4.5-Sonnet | **ECS (Ours)** | **$0.583 \pm 0.029$** | -- | -- |
> > | DeepSeek-V3.2 | Plain | $0.417 \pm 0.076$ | 0.000 | 1.000 |
> > | DeepSeek-V3.2 | Full Context | $0.417 \pm 0.058$ | 0.000 | 1.000 |
> > | DeepSeek-V3.2 | **ECS (Ours)** | **$0.417 \pm 0.076$** | -- | -- |
> >
> > In the main setting, ECS is higher than Plain LLM, Random, BM25, and SkillOpt on Pass^3 at p<0.05. Relative to Full Context, ECS is higher by 0.117, this comparison narrowly missed the conventional 0.05 threshold (p=0.058). We also revised the transfer discussion to report absolute results and mixed cases explicitly. With Claude-4.5-Sonnet, ECS obtains 0.678/0.617/0.583 on Pass^1/Pass^2/Pass^3, compared with 0.600/0.506/0.450 for Plain and 0.678/0.561/0.500 for Full Context. The Pass^3 comparisons against Plain and Full Context both give p=0.025. With DeepSeek-V3.2, ECS is slightly above Plain but below Full Context on Pass^1 (0.600 versus 0.583/0.633), below both on Pass^2 (0.472 versus 0.478/0.500), and tied with both on Pass^3 (0.417). We therefore describe transfer as model- and metric-dependent and avoid claiming uniformly positive transfer.
> >
> > > Clarify the distinction between selecting existing skill units and creating a new skill from a corpus.
> >
> > Thank you for highlighting this distinction. We now clarify in the main text, Section 3.1, that ECS searches over combinations of units in a preconstructed pool and is therefore largely extractive, whereas an authoring method can synthesize or compress corpus information into new skill units. We view these capabilities as complementary rather than claiming that ECS subsumes authoring.
> >
> > > The paper would also benefit from a clearer statement of its best applicability regime.
> >
> > Thank you for this suggestion. We updated Section 6, the limitations paragraph, to state explicitly that ECS is best suited to stable domains with a representative development set, a reliable automatic fitness metric, and sufficient artifact reuse to amortize the upfront search cost. We also clarify that rapidly changing domains or one-off settings may be better served by dynamic retrieval or direct authoring methods.